# TEST-TIME ADAPTATION WITH SLOT-CENTRIC MODELS

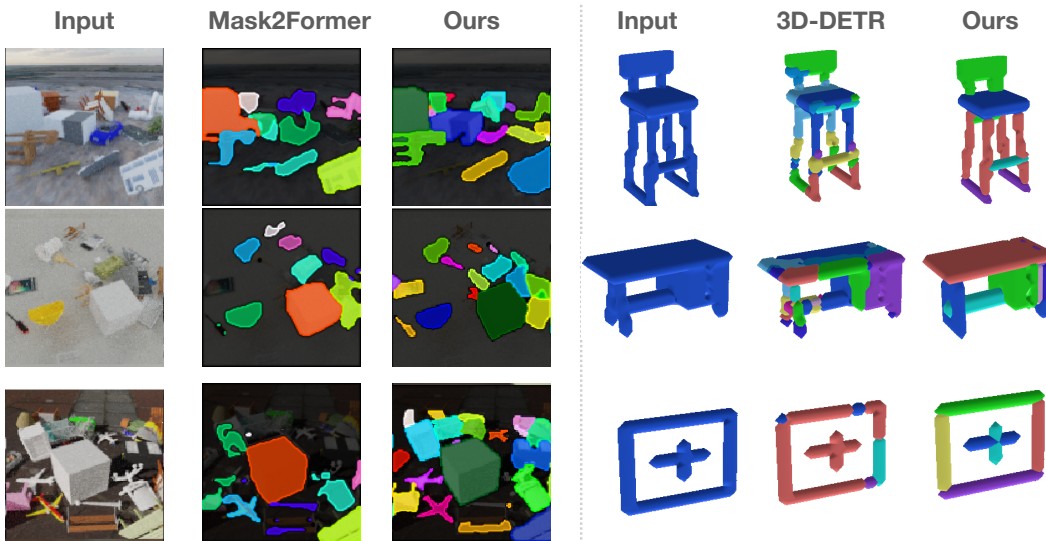

Figure 1: Image and point-cloud instance segmentation with **Slot-TTA.** Slot-TTA parse completely novel scenes into familiar entities via slow inference, i.e., gradient descent on the reconstruction error of the scene example under consideration. *Left:* Slot-TTA outperform Mask2Former (Cheng et al., 2021), a SOTA 2D image segmentor, on segmenting novel images by gradient descent on image synthesis of neighboring image views. *Right:* Slot-TTA outperform a state-of-the-art 3D-DETR detector by 30% in instance segmentation accuracy in out-of-distribution 3D point clouds, when trained on the same training data.

## ABSTRACT

We consider the problem of segmenting scenes into constituent objects and their parts. Current supervised visual detectors, though impressive within their training distribution, often fail to segment out-of-distribution scenes into their constituent entities. Recent test-time adaptation methods use auxiliary self-supervised losses to adapt the network parameters to each test example independently and have shown promising results towards generalization outside the training distribution for the task of image classification. In our work, we find evidence that these losses can be insufficient for instance segmentation tasks, without also considering architectural inductive biases. For image segmentation, recent slot-centric generative models break such dependence on supervision by attempting to segment scenes into entities in a self-supervised manner by reconstructing pixels. Drawing upon these two lines of work, we propose Slot-TTA, a semi-supervised instance segmentation model equipped with a slot-centric image or point-cloud rendering component, that is adapted per scene at test time through gradient descent on reconstruction or novel view synthesis objectives. We show that test-time adaptation greatly improves instance segmentation in out-of-distribution scenes. We evaluate Slot-TTA in several 3D and 2D scene instance segmentation benchmarks and show substantial out-of-distribution performance improvements against state-of-the-art supervised feed-forward detectors and self-supervised domain adaptation models.

## 1 INTRODUCTION

While significant progress has been made in machine scene perception and segmentation within the last decade, object (and part) detectors continue to generalize poorly outside their training distribution (Geirhos et al., 2020; Hendrycks et al., 2021). Consider the unfamiliar entity shown in Figure 1 (last row on the right). We can intuitively reason about meaningful parts that this shape could be broken into. Yet, a state-of-the-art 3D detection transformer (Misra et al., 2021) trained to segment chairs in a supervised manner struggles with this decomposition, even though the entitiy contains familiar (chair) parts. This lack of generalization requires us to build systems that can robustly adapt to such changes in distribution.

Test-time adaptation (TTA) (Ghifary et al., 2016; Sun et al., 2020; Wang et al., 2020) describes a setting where a model adapts to changes in distribution at test-time, at the cost of additional computation. In recent years, a variety of methods based on TTA have been proposed, focusing on few-shot adaptation (Ren et al., 2018) where the network is given access to a few labelled examples, or unsupervised domain adaptation (UDA) (Zhang, 2021) where the network is given access to many *unlabelled* examples from the new distribution. Of particular relevance is a specific UDA setting where model parameters are adapted *independently* to each unlabelled example in the test-set. This setting has been previously referred to as single-example UDA, and here we also refer to it as *slow inference* since it is similar to a human taking more time to parse a difficult example. Existing approaches for this setting typically devise a self-supervised loss that aligns well with the task of image classification and then optimize this loss during test-time adaptation (Sun et al., 2020; Gandelsman et al., 2022; Bartler et al., 2022; Grill et al., 2020). However, despite their success for image classification, these approaches do not provide adequate support for other scene understanding tasks, and in particular scene segmentation, as we showcase in Section 4.1.

One potentially important aspect to supporting TTA for other scene understanding tasks is the inductive bias of the underlying architecture. In the context of instance segmentation, there has been a lot of recent development in building models that segment scenes into entities in an unsupervised way by optimizing a reconstruction objective (Eslami et al., 2016; Greff et al., 2016; Van Steenkiste et al., 2018; Goyal et al., 2021; Du et al., 2020; Locatello et al., 2020; Zoran et al., 2021). These methods differ in details but share the notion of incorporating a fixed set of entities, also known as *slots* or *object files*. Each slot extracts information about a single entity during encoding, and is "synthesized" back to the input domain during decoding. Their ability to distinguish visual objects at a representation level makes them a particularly promising candidate for TTA for instance segmentation tasks.

In light of the above, we propose Test-time adaptation with slot-centric models (Slot-TTA), a semi-supervised slot-centric approach that combines Slot Attention (Locatello et al., 2020) (in the 2D image or point clouds setting) or Object Scene Representation Transformer (Sajjadi et al., 2022a) (in multi-view image setting) with a supervised segmentation loss to enable it to leverage instance-level image or point cloud annotations. Slot-TTA is trained jointly to synthesize and segment scenes. At test time, the model adapts without supervision to a single test sample by optimizing the self-supervised objective alone. Different from fully-unsupervised object-centric generative models, Slot-TTA uses annotations at training time to help it develop the notion of what an object is, which lets it scale to more complex visual settings. Different from existing TTA methods, Slot-TTA uses a slot-centric architecture and self-supervised synthesis loss that better aligns with the task of instance segmentation. Different from state-of-the-art detectors, Slot-TTA is equipped with reconstruction feedback that allows it to adapt at test time without supervision, i.e. without using additional annotated data. Indeed, we show that test-time adaptation via image or point cloud synthesis in Slot-TTA enables successfully parsing completely unfamiliar scenes composed of familiar entities (Figure 3).

We test Slot-TTA's instance segmentation ability on the following datasets: PartNet (Mo et al., 2019), MultiShapeNet-Hard (Sajjadi et al., 2022b) Multi-Shape and Plating. We evaluate Slot-TTA's ability to parse out-of-distribution scenes and compare it against state-of-the-art entity-centric generative models (Locatello et al., 2020; Sajjadi et al., 2022a), program synthesis models (Tian et al., 2019), 3D unsupervised part discovery models (Wu et al., 2020) and supervised visual detectors (Cheng et al., 2021; Misra et al., 2021) trained with labeled data to segment objects. We show improvements over all baselines in Slot-TTA ability to segment novel scenes. Additionally, we ablate different design choices of Slot-TTA. We will make our code and datasets publicly available to the community.

## 2 RELATED WORK

**Entity-centric generative models for scene decomposition**    *Entity-centric* (or *object-centric*) models use architectural inductive biases to represent perceptual inputs, such as an observation of a visual scene, in terms of separate object variables, often referred to as *slots* or *object files* (Greff et al., 2020; Sabour et al., 2017; Kosiorek et al., 2018; Engelcke et al., 2019; Goyal et al., 2020; Ke et al., 2021; Burgess et al., 2019; Greff et al., 2019; Zablotskaia et al., 2020; Rahaman et al., 2020). Prominent examples of such models include MONet (Burgess et al., 2019), GENESIS (Engelcke et al., 2019), IODINE (Greff et al., 2019), and Slot Attention (SA) (Locatello et al., 2020), which are trained in a fully-unsupervised setting via a simple auto-encoding objective. Object representations and scene decomposition emerge via the inductive bias of the model architecture (and in some cases, additional regularizers). However, without any form of supervision, scene decompositions can be ambiguous, which is particularly challenging for complex real-world scenes or in the presence of complicated textures. In Slot-TTA, we aid the competition mechanism in SA to address this issue by jointly training with a supervised segmentation loss. OSRT (Sajjadi et al., 2022a) is a cross-view geometry-free encoder-decoder method, that segments an image into objects through reconstructing novel viewpoints. OSRT combines SA with SRT (Sajjadi et al., 2022b), a view synthesis model that uses transformer encoder and decoders to fuse information across views, as well as the camera pose, without any explicit 3D information. Our multi-view RGB Slot-TTA builds upon their architecture.

**Test-time adaptation**    In test-time adaptation, model parameters are updated at test-time to better generalize to the distribution shift. In recent years, there has been significant development in this direction. Methods such as pseudo labelling and entropy minimization (Shin et al., 2022; Wang et al., 2020; Bateson et al., 2022) have demonstrated that supervising the model using its own confident predictions could help improve its accuracy. Adaptive BatchNorm methods (Khurana et al., 2021; Chang et al., 2019) have shown that updating the BatchNorm parameters using the new examples can help adaptation. Despite these successes, these methods by definition are data inefficient as they require confident predictions or a batch of examples to adapt. Self-supervised learning (SSL) (Sun et al., 2020; Bartler et al., 2022; Gandelsman et al., 2022) based methods on the other hand, have empirically shown to be data efficient. During training, they jointly train using the task and SSL loss, and during test-time, they train only using the SSL loss. All of the methods in the SSL setting thus far focus on the task of classification and mainly differ in terms of the SSL loss used. For example TTT (Sun et al., 2020) uses rotation angle prediction as their SSL loss, MT3 (Bartler et al., 2022) uses a BYOL (Grill et al., 2020) loss and TTT-MAE (Gandelsman et al., 2022) uses Masked autoencoding loss  (Pathak et al., 2016). In our work, we show that these losses do not generalize to segmentation, and how we might need specific architectural biases to close the gap.

We describe additional related work on **Unsupervised 3D Part Discovery** and **Shape program synthesis** in supplementary Section 10

## 3 METHOD

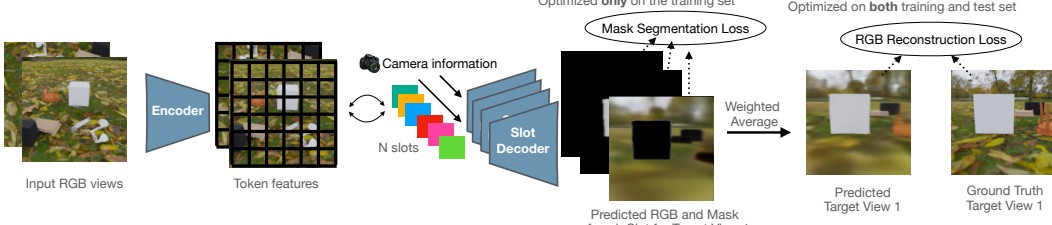

Figure 2: **Model architecture for multi-view images**. Given multi-view RGB images as input. Slot-TTA (here using OSRT (Sajjadi et al., 2022a) as a backbone) maps them to a set of token features, which are then mapped to a set of slot vectors. Conditioned on the camera-viewpoint Slot-TTA then decodes each slot into its respective segmentation mask and RGB image. It then uses weighted averaging to render the RGB image for the whole scene as seen from the camera viewpoint. On the training dataset, we jointly optimize using reconstruction and segmentation loss. On the test set, we optimize only using the reconstruction loss. We use a similar training pipeline for other input modalities.

The goal of Slot-TTA is to segment out-of-distribution scenes into objects and parts annotated in the training set. We consider three different settings: (i) segmenting 2D multi-view RGB images, (ii) 2D single-view RGB images, and (iii) segmenting 3D point clouds. In each setting, the model encodes the scene as a set of slot vectors (capturing information about individual objects), and decodes them back to either 3D point clouds or (novel-view) RGB images (depending on the setting). To compute slots, Slot-TTA uses Slot Attention (SA) (Locatello et al., 2020), where visual features are softly partitioned across slots through iterative attention. In the following, we first give a brief overview of SA in Section 3.1, followed by a detailed description of Slot-TTA in Section 3.2.

## 3.1 BACKGROUND

Current state-of-the-art detectors and segmentors instantiate slots (i.e. the query vectors) from 2D visual feature maps or 3D point feature clouds (Carion et al., 2020; Misra et al., 2021). Most works use iterative cross-attention (features to slots) and self-attention (slot-to-slots) operations (Carion et al., 2020) to map a set of $N$ input feature vectors to a set of $K$ slot vectors. Attention-based competition amongst slots and iterative routing popularized in Goyal et al. (2021); Locatello et al. (2020) encourages a single location in the input to be assigned to a unique slot vector.

Given a visual scene encoded as a set of feature vectors $M \in \mathbb{R}^{N \times C}$ and $K$ randomly initialized slots sampled from a multivariate Gaussian distribution with a diagonal covariance $S \sim \mathcal{N}(\mu, \text{Diag}(\sigma^2)) \in \mathbb{R}^{K \times D}$, where $\mu, \sigma \in \mathbb{R}^C$ are learnable parameters of the Gaussian, Slot Attention (Locatello et al., 2020) computes an attention map $a$ between the feature map $M$ and the slots $S$:

$$a = \text{Softmax}(k(M) \cdot q(S)^T, \text{axis=``slots''}) \in \mathbb{R}^{N \times K}. \tag{1}$$

$k$, $q$, and $v$ are learnable linear transformations that map inputs and slots to a common dimension $D$. The softmax normalization over slots ensures competition amongst them to attend to a specific feature vector in $M$. Updates to the slots are computed based on the input features they attend to:

$$updates = a^T v(M) \in \mathbb{R}^{K \times C}, \text{ where } a_{i,k} = \frac{a_{i,k}}{\sum_{i=0}^{N-1} a_{i,k}} \tag{2}$$

which are then fed into a GRU (Cho et al., 2014): $S = \text{GRU}(\text{state} = S, \text{input} = updates)$. We iterate 3 times over equations 1 and 2. For detailed description, please refer to Locatello et al. (2020).

## 3.2 TEST-TIME ADAPTATION WITH SLOT-CENTRIC MODELS (SLOT-TTA)

We first describe the encoders and decoders that form the foundation of Slot-TTA for each modality. Further we detail how we train Slot-TTA and perform test time adaptation through slow inference.

### 3.2.1 ENCODING AND DECODING BACKBONES

**Posed multi-view 2D RGB images** As shown in Figure 2, Slot-TTA builds upon the architecture of OSRT (Sajjadi et al., 2022a), which is an object-centric, geometry-free novel view synthesis method. Given a set of multi-view RGB images as input, a CNN encodes each input image $I_i$ into a feature grid, which is then flattened into a set of tokens with camera pose and ray direction information added in each of the tokens, similar to SRT (Sajjadi et al., 2022b). These are then encoded into a set of latent features using a transformer (Vaswani et al., 2017) $\text{Enc}$ with multiple self-attention blocks $z = \text{Enc}(\text{CNN}(I_i))$. The latent features $z$ are then mapped into a set of slots $S$ using Slot Attention (Section 3.1). For decoding, we adopt the spatial broadcast decoder (Watters et al., 2019) formulation, where a render MLP takes as input the slot vector $S_k$ and the pixel location $p$ parameterized by the camera position and the ray direction pointing to the pixel to be decoded. It outputs an RGB color $c_k$ and an unnormalized alpha score $a_k$ for each pixel location $c_k, a_k = \text{Dec}(p, S_k)$. The $a_k$'s are normalized using a Softmax and used as weights to aggregate the predicted RGB values $c_k$ for each slot. A camera viewpoint conditioned decoder allows us to render novel viewpoints, for which we show novel view rendering results in our supplementary video. We ablate other decoder choices, such as the Slot Mixer decoder (Sajjadi et al., 2022a) in supplementary Section 9.1.

**Single-view 2D RGB images** For this setting, Slot-TTA uses a ResNet-18 (He et al., 2016) to encode the input RGB image into a feature grid. We then add positional vectors to the feature grid and map to a set of slot vectors using Slot Attention. Similar to the multi-view setting, each slot

vector is decoded to the RGB image and an alpha mask using an MLP renderer. We parameterize pixel location $p$ as $(x, y)$ points on the grid instead of camera position as the above setting.

**3D point clouds** To adapt to 3D point clouds, Slot-TTA uses a 3D point transformer (Zhao et al., 2021) which maps the 3D input points to a set of $M$ feature vectors of $C$ dimensions each. We set $M$ to 128 and $C$ to 64 in our experiments. Point feature vectors are mapped to slots with Slot Attention. Slot-TTA decodes 3D point clouds from each slot using implicit functions (Mescheder et al., 2019). Specifically, each decoder takes in as input the slot vector $S_k$ and an $(X, Y, Z)$ location and returns the corresponding occupancy score $o_{k,x,y,z} = \text{Dec}(S_k, (x, y, z))$, where Dec is a multi-block ResNet MLP similar to that of Lal et al. (2021). We then max-pool over the slot dimension $k$ to get an occupancy value $o_{x,y,z}$ for each 3D point in the scene.

### 3.2.2 TRAINING AND TEST-TIME ADAPTATION

Slot-TTA assumes entity-level supervision in the form of segmentation masks, and can also exploit unlabelled data via a reconstruction loss objective.

**Training for joint segmentation and reconstruction** We train all the parameters of our model to jointly optimize image / point cloud reconstruction or novel view image synthesis objectives and the task segmentation objective over all the $n$ examples in the training set, where $x$ represents the input scene and $y$ the segmentation labels:

$$\min_\theta \frac{1}{n} \sum_{i=1}^{n} \lambda_s l_{seg}(x_i, y_i; \theta) + \lambda_r l_{recon}(x_i; \theta) \tag{3}$$

In the case of RGB images, for reconstruction, we minimize the mean squared error between the predicted and ground truth RGB images. For segmentation, we supervise the alpha masks $a_i$ of each slot as provided by the decoders. We use Hungarian matching (Kuhn, 1955) (combinatorial optimization algorithm that solves assignment problems) to associate the ground truth masks with the predicted masks, and upon association we apply a categorical cross-entropy loss $l_{seg}$. In the case of 3D point clouds we supervise the predicted occupancy probability $o_k$. We use a binary cross-entropy loss for $l_{recon}$. For $l_{seg}$ we use Hungarian matching with a categorical cross-entropy loss similar to other modalities. We weight the segmentation and reconstruction loss by $\lambda_s$ and $\lambda_r$.

**Test-time adaptation** In this work, we refer to a single forward pass through our trained model without any test-time adaptation as *fast inference* (same as regular inference). We call the process of test-time adapting the model on each example independently *slow inference*, using only the reconstruction objective of Eq. 3. We use this terminology to emphasize that the only difference between both settings is the added computation time which results in an effective speed difference between the two inference schemes. We adapt only the encoder parameters $\theta_{enc}$ in our model, which we found to improve results compared to adapting the entire model as shown in our supplementary Section 9.1. We train for 150 steps per example using the Adam optimizer (Kingma & Ba, 2014) and a learning rate of 1e-4.

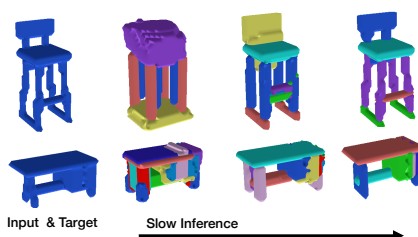

Figure 3: **Slow Inference (TTA) in Slot-TTA**: 3D segmentation improves over gradient steps despite only optimizing for 3D reconstruction.

## 4 EXPERIMENTS

We test Slot-TTA capability for segmenting posed multi-view RGB images, single-view RGB images and 3D point clouds. Our experiments aim to answer the following questions: (i) How does Slot-TTA compare against state-of-the-art 2D and 3D segmentation models (Luo et al., 2020; Yu et al., 2021; Cheng et al., 2021)? (ii) How does slow inference through reconstruction feedback affect segmentation accuracy in Slot-TTA and its variants? (iii) How much does supervision during training contribute to segmentation performance?

**Evaluation metric**   In addition to the loss, we use Adjusted Random Index (ARI) as our evaluation metric for segmentation accuracy (Rand, 1971). ARI calculates the similarity between two-point clusters while being invariant to the ordering of the cluster centers. For this, we use the publicly available implementation of Kabra et al. (2019).

### 4.1   SEGMENTING RGB IMAGES IN MULTI-VIEW SCENES

**Dataset**   We evaluate Slot-TTA on the MultiShapeNet (MSN) dataset from SRT (Sajjadi et al., 2022b). The dataset is constructed by rendering 51K ShapeNet objects using Kubric (Greff et al., 2022) against 382 photo-realistic HDR backgrounds so that there is no overlap of objects between the train and test sets. In addition to having different object instances in training and test sets, we further re-generate data in the MSN dataset so that train and tests sets differ in the number of objects present: scenes with 5-7 object instances are in the training set and scenes with 16-30 objects are in the test set. Each instance is sampled from a random pose. This lets us measure how well our model and the baselines perform under this distribution shift. Please refer to supplementary Section 7 for further dataset details and visualization of samples from the train-test split. Additionally in Table 6, we test our model on a different distribution shift, where instead of increasing the number of instances in the test-set we introduce new object categories from Google Scanned objects dataset Downs et al. (2022). Thus showing Slot-TTA improves performance across different distribution shifts.

| Method | in-dist (5-7 instances) | | out-of-dist (16-30 instances) | |
|---|---|---|---|---|
| | Fast Infer. | Slow Infer. | Fast Infer. | Slow Infer. |
| Slot-TTA-w/o supervision | 0.32 | 0.30 | 0.33 | 0.29 |
| Mask2Former | **0.93** | N/A | 0.74 | N/A |
| Mask2Former+BYOL | **0.93** | **0.95** | **0.75** | 0.74 |
| Mask2Former+Recon | **0.93** | 0.92 | 0.74 | 0.67 |
| Slot-TTA (Ours) | 0.92 | **0.95** | 0.70 | **0.83** |

Table 1: **Instance Segmentation accuracy** (higher is better) in the multi-view RGB setup for in-distribution test set of 5-7 object instances and out-of-distribution 16-30 object instances.

**Baselines**   We compare to three baselines: (i) Mask2Former (Cheng et al., 2021), a state-of-the-art 2D image segmentor which adapts detection transformers (Carion et al., 2020) to image segmentation by using multiscale segmentation decoders with masked attention. (ii) Mask2Former+BYOL which combines the segmentation model of (Cheng et al., 2021) with test time adaptation using BYOL self-supervised loss of Bartler et al. (2022). (iii) Mask2Former+Recon which combines the segmentation model of Cheng et al. (2021) with rendering submodules and image reconstruction loss for test-time adaptation.

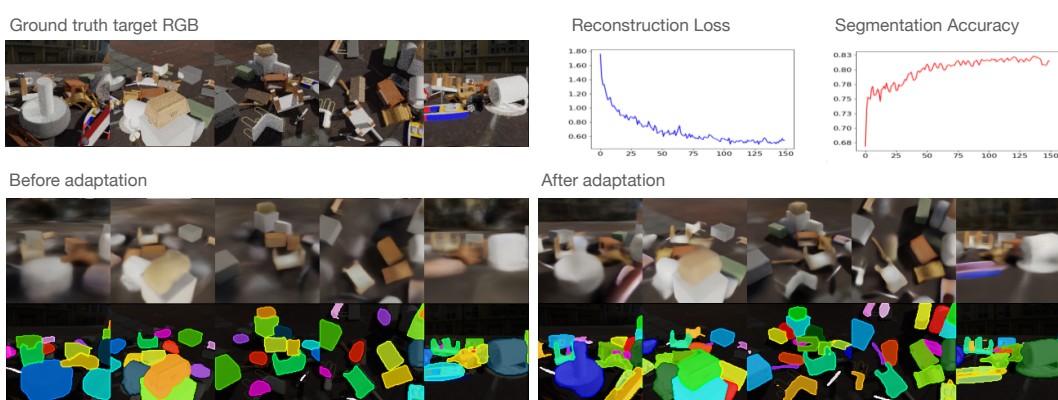

Figure 4: **Test-time adaptation via slow inference in Slot-TTA for multi-view scenes**. In the right top we visualize the RGB loss (blue curve) and the segmentation ARI accuracy (red curve). As can be seen, during slow inference the segmentation accuracy improves as reconstruction loss reduces.

**Results** We show quantitative segmentation results of our model and baselines on target camera viewpoints in Table 1 and qualitative TTA results in Figure 4. In Slot-TTA-w/o supervision, instead of training jointly for reconstruction and segmentation, we train using only cross-view image synthesis, similar to OSRT (Sajjadi et al., 2022a).

It can be observed that: (i) Slot-TTA-Slow outperforms the feedforward Mask2Former-Fast, especially for out-of-distribution scenes; (ii) adding self-supervised losses of SOTA image classification methods (Bartler et al., 2022) to Mask2Former (eg. Mask2Former+BYOL) does not suffice to adapt them effectively at test time and (iii) Slot-TTA without supervision, which is identical to OSRT is not competitive with supervised models for object segmentation.[1]

For additional qualitative comparisons between fast and slow inference in the multi-view setting, please refer to Figure 11 in Supplementary. For extensive ablations of Slot-TTA please refer to Table 5 in Supplementary. For novel-view renderings please refer to our supplementary video.

## 4.2 SEGMENTING SINGLE-VIEW RGB IMAGES

As a proof of concept, in this section, we test our model and the baseline Mask2Former in segmenting single RGB images comprised of multiple samples from five shapes of distinct colors, organized in heavily occluded configurations, a dataset we create and we call Multi-Shape. Our training set consists of images with 3-5 object instances, while the test set consists of images with 10-16 object instances. For this setting, we report the ARI scores for the foreground objects only, since in this dataset the background occupies a large image area and a method that assigns most pixels to background already achieves a very high ARI. We find the performance accuracy ordering of the methods to be the same.

As can be seen in Table 2, before TTA Mask2former and Mask2former+Recon outperform our method. After TTA, our method significantly outperforms the baselines.

| Method | in-dist (3-5 instances) | | out-of-dist(10-16 instances) | |
|---|---|---|---|---|
| | Fast Infer. | Slow Infer. | Fast Infer. | Slow Infer. |
| Mask2Former | **0.96** | N/A | **0.44** | N/A |
| Mask2Former+Recon | 0.95 | 0.94 | 0.43 | 0.47 |
| Slot-TTA (Ours) | **0.96** | **0.95** | 0.39 | **0.69** |

Table 2: **Foreground instance segmentation accuracy** (higher is better) for single-view RGB images. In-distribution images have 3-5 objects and out-of-distribution images have 10-16 objects.

Please refer to Section 9.2 for qualitative results in our Multi-Shape dataset, where we showcase some success and failure cases in Slot-TTA with slow inference.

Additionally we test Slot-TTA on a real-world salad plating dataset which we collected and will publicly release. As can be viewed at [https://sites.google.com/corp/view/slottta], our model effectively segments the scene into objects and amodally reconstructs each one, despite heavy occlusions.

## 4.3 SEGMENTING 3D POINT CLOUDS

We test Slot-TTA in segmenting 3D object point clouds into parts, for within distribution and out-of-distribution object categories.

We consider two segmentation supervision setups: (i) Supervision from a dataset of generic 3D part primitives. (ii) Supervision from labelled 3D object point-clouds of a related object category.

### 4.3.1 SUPERVISION FROM A DATASET OF GENERIC 3D PART PRIMITIVES

**Dataset** We use the part primitive dataset introduced by Shape2Prog (Tian et al., 2019) (akin to generalized cylinders of Marr (1982)), which consists of differently sized cubes, cuboids, and discs. Thus, our supervised training set consists of scenes that contain a single primitive part, resized and

---

[1]Although OSRT performs poorly in the ARI metric, it achieves substantially better results in terms of foreground-ARI (yet still not competitive). This is because it is unable to segment out the background.

translated in different 3D locations of a blank 3D canvas, while the test set consists of unseen chairs and tables from ShapeNet, each composed of 6 to 16 parts. Our goal is to quantify to what extent our model can compose generic parts into shape compositions via slow inference, as we motivated in Figure 1. We assume access to unlabelled 3D chair point clouds during joint training.

**Baselines** We compare our model against the following baselines: (i) PQ-Nets of Wu et al. (2020) assume access to a set of primitive 3D parts for pre-training. Specifically they first learn a primitive part decoder, then they learn a sequential encoder that encodes the 3D point cloud into a 1D latent vector and sequentially decodes parts using the pretrained part decoder. We use the publicly available code to train the model. (ii) Shape2Prog of Tian et al. (2019) is a shape program synthesis method that is trained supervised to predict shape programs from object 3D point clouds. The program represents the part category, location, and the symmetry relations among the parts (if any).

We further evaluate Slot-TTA w/o supervision, an ablative version of our model that is trained without any supervised pre-training on the generic part dataset, but is only trained using a reconstruction objective for autoencoding the Chair dataset. This version of our model coincides with the previous work of Slot Attention of Locatello et al. (2020) but instead implemented for 3D.

| Method | in-dist (Chairs) | | out-of-dist (Tables) | |
|---|---|---|---|---|
| | Fast Infer. | Slow Infer. | Fast Infer. | Slow Infer. |
| Shape2Prog (Tian et al., 2019) | 0.28 | 0.53 | 0.23 | 0.40 |
| PQ-Nets (Wu et al., 2020) | 0.20 | 0.31 | 0.17 | 0.21 |
| Slot-TTA w/o supervision | 0.41 | 0.35 | 0.47 | 0.38 |
| Slot-TTA (Ours) | **0.57** | **0.62** | **0.60** | **0.69** |

Table 3: **Instance Segmentation accuracy** (higher is better) in the test set of Chair category (in-distribution) and Table category (out-of-distribution) when trained using the supervision of generic primitives.

**Results** We show quantitative results of our model and the baselines in Table 3. It can be observed that: (i) Slot-TTA significantly outperform PQ-Nets (Wu et al., 2020), which maps the input object 3D pointcloud into a 1D latent vector, suggesting that segregation into slot like entities using attention as in Slot-TTA is beneficial; (ii) Slot-TTA-Fast outperforms Slot-TTA w/o supervision-Fast by a large margin, indicating that the additional supervised data is beneficial and correctly integrated; and (iii) slow inference through reconstruction feedback helps in the presence of supervision and hurts in the absence of it. Such trade-off between reconstruction and segmentation in generative models for scene decomposition has previously been pointed out in Engelcke et al. (2020), which is also supported by our findings.

In supplementary, please, refer to Section 9.3.1for further ablations and qualitative comparison against baselines. Refer to the supplementary video for the intermediate visualizations of slow inference. Finally please refer to Figure 7.3.1 for visualization of the primitive dataset.

### 4.3.2 SUPERVISION FROM A RELATED OBJECT CATEGORY

**Dataset** In this setup, we test our model and baselines in segmenting the test objects of the Chair and Table categories in the PartNet benchmark (Mo et al., 2019), with access to ground-truth pointcloud segmentation of the Chair category in PartNet. We train Slot-TTA in a semi-supervised way combining a reconstruction loss and a supervised segmentation loss as described in Section 3.2.2.

**Baselines** We compare our model against the following baselines: (i) Learning2Group of (Luo et al., 2020) progressively groups points into segments by learning pairwise grouping decisions parameterized by features of the two point clusters. We used the publicly available code and trained the model in the training set of the Chair category. (ii) 3D-DETR a variant of state-of-the-art 3D object detection model of Misra et al. (2021) for 3D point-cloud instance segmentation. Please refer to supplementary Section 8.4 for a detailed descriptions on all the baselines.

We consider the following ablative versions of Slot-TTA: (i) Slot-TTA w/o supervision is trained without the supervised part segmentation loss; and (ii) Slot-TTA w/o SlotAttention, which does not

| Method | in-dist (Chair) | | out-dist (Table) | |
|---|---|---|---|---|
| | Fast Infer. | Slow Infer. | Fast Infer. | Slow Infer. |
| 3D-DETR (Misra et al., 2021) | **0.67** | N/A | 0.41 | N/A |
| Learning2Group (Luo et al., 2020) | 0.62 | N/A | 0.46 | N/A |
| Slot-TTA w/o supervision | 0.40 | 0.44 | 0.31 | 0.48 |
| Slot-TTA w/o SlotAttention | 0.58 | 0.55 | 0.31 | 0.44 |
| Slot-TTA (Ours) | 0.64 | **0.66** | **0.49** | **0.61** |

Table 4: **Instance Segmentation accuracy** (higher is better) in the test set of Chair category (in-distribution) and Table category (out-of-distribution) when trained using the supervision of Chair category.

use Slot Attention for mapping point features to slots. Instead it maps 3D point features to slots via iterative layers of cross (query to point) and self (query-to-query) attention layers on learnable query vectors similar to 3D-DETR and DETR (Carion et al., 2020). Please note that slots and queries represent the same thing, but we use the terminology of DETR (Carion et al., 2020) in this case.

**Results** We report the fast and slow inference results for all ablative versions of our model. Our baselines in this case, 3D-DETR and Learning2Group (Luo et al., 2020) are feedforward in nature, they are not equipped with decoders, and thus cannot be evaluated with slow inference. We show quantitative results in Table 4 and qualitative results in Figure 5. Slot-TTA significantly outperform the baselines, and Slot-TTA-Slow results in a significant boost in performance ($\sim 30\%$) over the feedforward inference in our model, Slot-TTA-Fast.

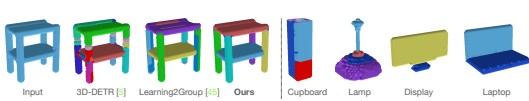

Figure 5: **Out-of-distribution 3D segmentation results** for Slot-TTA and baselines. *Left:* 3D segmentation results on out-of-distribution shapes for Slot-TTA and baselines. *Right:* 3D segmentation results for out-of-distribution categories of PartNet for Slot-TTA trained semi-supervised only on the Chair category.

We draw the following conclusions from Table 4: (i) Generic primitives generalize a lot better than category-specific supervision. Slot-TTA in Table 3 (that uses the generic parts dataset) outperforms Slot-TTA in Table 4 (that use the category-specific supervision) in OOD generalization on the Table category. (ii) Competition amongst slots helps. Slot-TTA outperforms Slot-TTA w/o SlotAttention, thus showing competition amongst slot vectors during encoding helps generalization. (iii) Slow inference through reconstruction feedback helps OOD generalization of Slot-TTA. Our baselines 3D-DETR and Learning2Group (Luo et al., 2020) are feed-forward in nature, they lack any form of reconstruction feedback, and thus cannot adapt as our model through such feedback. In our supplementary Section 9.3.2 we extend Table 4 *to all remaining 14 ShapeNet categories* as the test categories. Further in Table 8 in supplementary we report results on bounding box detection instead of instance segmentation, and showcase a similar improvement in performance. Thus showcasing our method generalizes beyond segmentation tasks.

## 5 CONCLUSION

We presented Slot-TTA, a novel semi-supervised instance segmentation model equipped with a slot-centric image or point-cloud rendering component. Slot-TTA is capable of test-time adaptation on a single unlabeled example (i.e. slow inference) through gradient descent on reconstruction or novel view synthesis objectives. We have shown how slow inference for Slot-TTA greatly improves segmentation in out-of-distribution scenes, and compares favorably to other (unsupervised) segmentation approaches, including other forms of test time adaptation.

There exist several promising directions for improving Slot-TTA, which reflect its current limitations. Currently, Slot-TTA does not model pairwise interactions between slots, while such cross-talk could be beneficial for adaptation. Further, while a scene decomposition into entities and parts is inherently hierarchical, such structure is currently not reflected in Slot-TTA as the backbones it considers capture a flat, non-hierarchical, list of entities.

## 6 REPRODUCIBILITY STATEMENT

To ensure the reproducibility of our work, we will make our code and datasets publicly available to the community. Additionally, in our supplementary section, we try our best to mention all the information that could aid reproducibility. Specifically, in Section 7 we mention all the dataset preparation details. In Section 8 we in-depth specify the implementation details of our method and the baselines, including the hyperparameter values and computational details.

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

APPENDIX

The structure of this appendix is as follows: In Section 7 we cover the details on the datasets. In Section 8 we specify further implementation details. In Section 9 we provide additional qualitative and quantitative results for the experiments in Section 4 of our main paper.

Along with this we also provide a video file in the supplementary zip, which visualizes the intermediate reconstruction and parsing results of Slot-TTA during the slow inference stage and we also visualize our multi-view rendering results.

## 7 DATASETS

### 7.1 MULTI-VIEW RGB

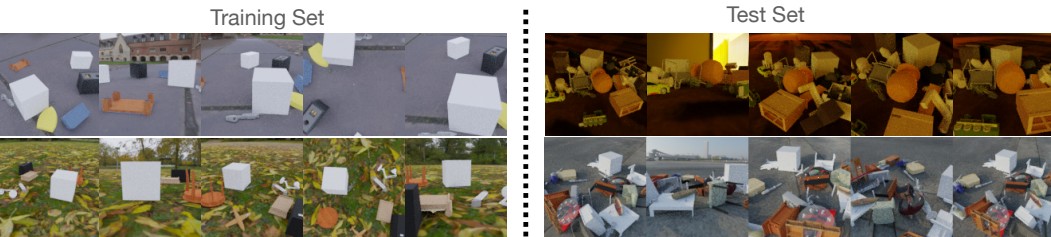

Figure 6: We visualize samples from the train-test split used by us in experiment Section 4.1. Different rows correspond to different scenes and different columns correspond to different viewpoints.

We use the MultiShapeNet-Hard dataset of Scene Representation Transformer, a complex photo-realistic dataset for Novel View Synthesis (Sajjadi et al., 2022b). Our train split consists of 5-7 ShapeNet objects placed at random locations and orientations in the scene. The backgrounds are sampled from 382 realistic HDR environment maps. Our test set consists of 16-30 objects placed at novel arrangements. We sample objects from a pool of 51K ShapeNet objects across all categories, we divide the pool into train and test such that the test set consists of objects not seen during training. The train split has 200K scenes, and the test set consists of 4000 scenes, each with 10 views. We had to regenerate the dataset for this specific train-test split.

### 7.2 SINGLE-VIEW RGB

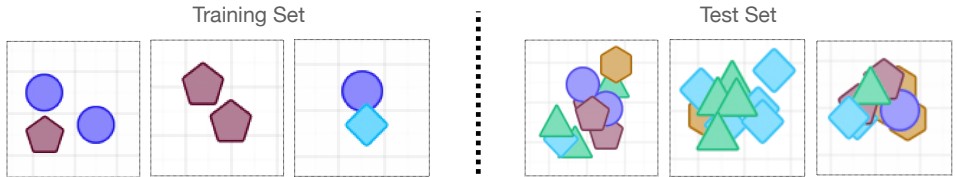

Figure 7: We visualize the samples of our MultiShape dataset.

Multi-Shape is a dataset built by us for proof-of-concept. It consists of 5 shapes of distinct colors uniformly placed at a random location in a 2D canvas. Our training set consist of 3-5 object instances, while the test set consists of a highly occluded setting with 10-16 object instances.

### 7.3 POINT CLOUD

For all the tasks, we subsample the input point clouds to a standard size of 2048 points.

#### 7.3.1 GENERIC PRIMITIVE PART DATASET.

We use the primitive dataset of Tian et al. (2019) as supervision in Experiment Section 4.3.1. The dataset consists of 200K primitive instances sampled from the primitive templates that are visualized

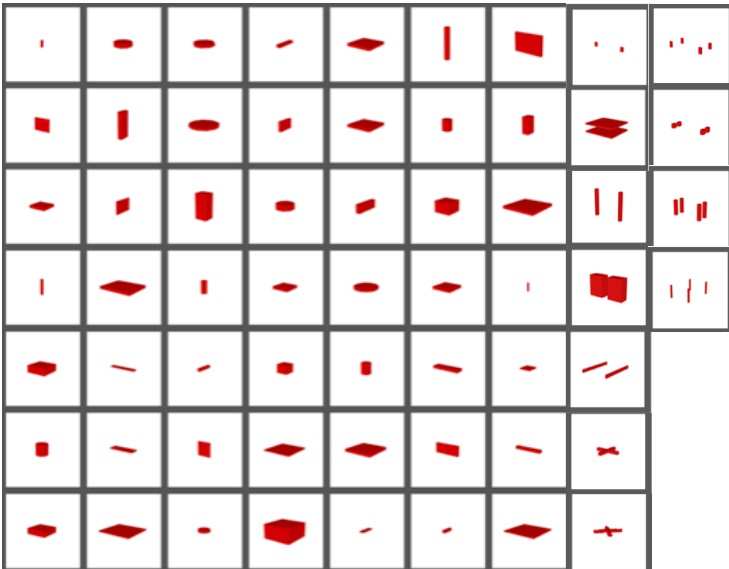

Figure 8: We visualize all the generic primitive templates of Tian et al. (2019), as you can see, they mainly consist of Cubes, Cuboids, and Discs.

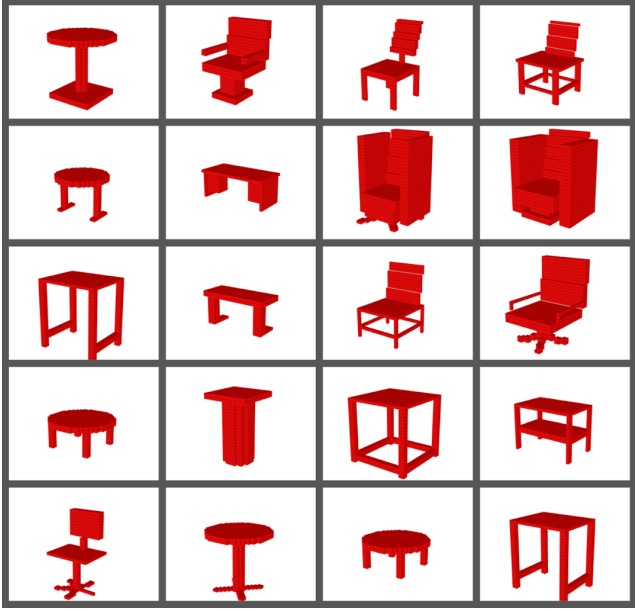

Figure 9: We visualize the Synthetic whole shape dataset of Tian et al. (2019). Shape2Prog supervises their model using the annotations from this dataset.

in Figure 8. Examples are sampled from the templates by changing their sizes and placing them uniformly in random locations, similar to Tian et al. (2019).

### 7.3.2 SYNTHETIC WHOLE SHAPE DATASET.

This dataset was generated by Shape2Prog (Tian et al., 2019). The segmentation labels from this dataset are used by them as a supervision signal for later generalizing to PartNet shapes. The dataset consists of about 120K synthetically generated Chairs and Tables, we visualize some of these synthetically generated tables and chairs in Figure 9. Note that neither Slot-TTA nor the baselines have access to this dataset.

### 7.3.3 PartNet dataset.

We use the official level-3 train-test split of PartNet (Mo et al., 2019). We use the train split of Chair category as our training set, we consider test split of Table category in PartNet our test categories. We use this as the train-test split in Experiment Section 4.3.2. We set the value of number of slots $K$ as 16 for this dataset.

## 8 Implementation details

### 8.1 Posed multi-view 2D RGB images

**Training details and computational complexity.** We use a batch size of 256 in this setting. We set our learning rate as $10^{-4}$. We use an Adam optimizer with $\beta_1 = 0.9$, $\beta_2 = 0.999$. For training, our model takes about 4 days to converge using 64 TPUv2 chips. Our slow inference for each example takes about 10 seconds on a single TPUv2 chip. Similarly, a forward pass through our model takes about 0.1 seconds. During training, instead of decoding all the pixels, we decode only a sample of them. Specifically, we randomly pick 1024 pixel locations for each example in the batch during each iteration of training. During test-time adaptation, instead of uniformly sampling pixel locations, we use an error-weighted sampling strategy which we describe below.

**Inputs.** During training and test-time adaptation, our model takes in as input multi-view RGB images along with their ground-truth egomotion. For each scene, we randomly select 5 input and target views, and make sure there is no overlap between the two sets, as a result rendering novel viewpoints each time. Note that although we stick to 5 viewpoints, Slot-TTA can take a variable number of viewpoints as input. We use a resolution of 128x128 for our input and target images.

**Encoder.** Here we follow the original implementation of OSRT (Sajjadi et al., 2022b). The model encodes each input image $I_i$, its camera extrinsic and intrinsics into a set representation via a shared CNN and transformer backbone. Specifically, the CNN outputs a feature grid for each image conditioned on the camera extrinsic and intrinsics, which are then flattened into a set of flat patch embeddings. The patch embeddings are then processed by a transformer that outputs a set of latent embeddings. The latent embeddings have a dimensionality of 1535. The CNN consists of 3 blocks of convolutions, with a ReLU activation after each convolution. The transformer contains 5 blocks of Multi-Head Self-attention.

**Slot Attention.** The latent embeddings from the encoder are then mapped to a Slot Attention module. We use the original implementation by Locatello et al. (2020), however instead of initializing the slots from a multi-variate gaussian we have them as learnable embedding vectors. We keep our slot vectors dimensionality as 1536. We set the number of slots as 32 in this setting.

**Decoder.** We use the broadcast decoder of Sajjadi et al. (2022a) for decoding the slots to their RGB image conditioned on the target viewpoints. Our slot decoder consists of a 4-layer MLP with a hidden dimensionality of 1536 and ReLU activation. Our target viewpoints are parameterized using 6D light-field parametrization of camera position and normalized ray direction.

**Error-conditioned pixel sampling** To accelerate test-time adaptation, we sparsely sample a subset of pixels from the target images, where we prioritize the pixels with a high reconstruction error. To this end, we calculate the reconstruction error over all pixels and apply a Softmax with a temperature $\tau = 0.01$ along the pixel dimension.

### 8.2 Single-view 2D RGB images

**Training details and computational complexity.** We use a batch size of 16 in this setting. We set our learning rate as $10^{-4}$. We use an Adam optimizer with $\beta_1 = 0.9$, $\beta_2 = 0.999$. For training, our model takes about 12 hours to converge using V100 GPU. Our slow inference for each example takes about 6 seconds on the same GPU. Similarly, a forward pass through our model takes about 0.06 seconds. We decode the whole image instead of sampling a subset of pixels like in the above section.

**Inputs.**    We use a single 2D RGB image of resolution 128 x 128 as our input. We normalize our input in the range of -0.5 to 0.5 before passing it through our encoder.

**Encoder.**    We use ResNet-18 (He et al., 2016) as our encoder backbone, which takes as input the RGB image and outputs a feature grid. We add positional vectors to the feature grid, which are grid locations normalized in the range of -1.0 to -0.1.

**Slot Attention.**    We use the exact implementation of Locatello et al. (2020). In this setting, we set the number of slots as 16 as that's the maximum number of objects in Multi-Shape dataset.

**Decoder.**    We use a 4-layer MLP decoder, that takes in as input a slot vector and a 2D location and outputs it's corresponding RGB value and alpha score.

## 8.3    3D POINT CLOUDS

**Training details and computational complexity.**    We use a batch size of 8 for point cloud input. We set our learning rate as $40^{-4}$. We use the Adam optimizer with $\beta_1 = 0.9$, $\beta_2 = 0.999$. Our model takes 24 hours (approximately 200k iterations) to converge. Our slow inference per example takes about 1 min (500 iterations). A forward pass through the proposed model takes about 0.15 secs. We use a single V100 GPU for training and inference.

**Inputs.**    We subsample the input point clouds to a standard size of 2048 points.

**Encoder.**    We adopt the point transformer (Zhao et al., 2021) architecture as our encoder. Point transformer encoder is essentially layers of self attention blocks. Specifically a self attention block includes sampling of query points and updating them using their $N$ most neighbouring points as key/value vectors. In the architecture we specifically apply 5 layers of self attention which look as follows: 2048-16-64, 2048-16-64, 512-16-64, 512-16-64, 128-16-64, 128-16-64. We use the notation of $S$-$N$-$C$, where $S$ is the number of subsampled query points from the point cloud, $N$ is the number of neighbouring points and $C$ is the feature dimension. We thus get an output feature map of size $128 \times 64$.

**Decoder.**    We obtain point occupancies by querying the slot feature vector $slot_k$ at discrete locations $(x, y, z)$ specifically $o_{x,y,z} = \text{Dec}(slot_k, (x, y, z))$. The architecture of $\text{Dec}$ is similar to that of Lal et al. (2021). Given $slot_k$, which is one of the slot feature vector. We encode the coordinate $(x, y, z)$ into a 64-D feature vector using a linear layer. We denote this vector as $z$. The inputs $slot_k$ and $z$ are then processed as follows:

$$out_k = RB_i(RB_{i-1}(\cdots RB_1(z + FC_1(slot_k)) \cdots) + FC_{i-1}(slot_k)) + FC_i(slot_k). \quad (4)$$

We set $i = 3$. $FC_i$ is a linear layer that outputs a 64 dimensional vector. $RN_i$ is a 2 layer ResNet MLP block (He et al., 2016). The architecture of ResNet block is: ReLU, 64-64, ReLU, 64-64. Here, $i - o$ represents a linear layer, where $i$ and $o$ are the input and output dimension. Finally $out_k$ is then passed through a ReLU activation function followed by a linear layer to generate a single value for occupancy.

## 8.4    BASELINES

**Mask2former** (Cheng et al., 2021)    Mask2former is a recent state-of-the-art 2D RGB segmentation network, that scales transformer-based 2D-DETR (Carion et al., 2020) for the task of segmentation. They improve 2D-DETR's transformer decoder by adding masked and multi-scale attention, which helps them achieve SOTA results on panoptic, instance and semantic segmentation on the COCO dataset. We use their publicly available code to train on MultiShapeNet dataset. We use a batch size of 256 and train their network on 8 V100s GPUS for 4 days until convergence. We set the number of slots in their network as 32, similar to our model.

**3D-DETR.**    3D-DETR is a version of 3DETR (Misra et al., 2021)(a 3D state-of-the-art object detection method) scaled to the task of instance segmentation, we build this architecture on top of the idea of 2D-DETR (Carion et al., 2020).

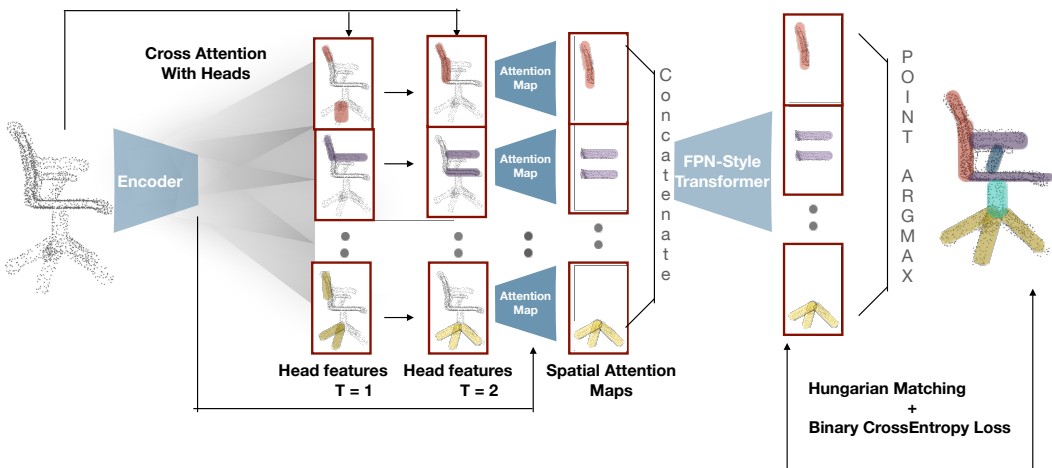

Figure 10: Given point clouds as input the encoder backbone featurizes the points into $N$ feature vectors, we then do iterative self and cross attention using $K$ learnable query heads, similar to Carion et al. (2020). We then compute attention of the updated queries wrt to the encoded feature vectors. We then concatenate these attentions along the batch dimension and pass them through a FPN-style transformer that increases their resolution and outputs each query mask logits. We then do hungarian matching and binary cross entropy loss

The base encoder is a 5 layer architecture of 1024-16-64, 1024-16-64, 1024-16-64, 1024-16-64, 512-16-64, following the notation of $S$,$N$,$C$ where $S$ is the number of sampled query points, $N$ is the number of selected neighbouring points and $C$ is the feature dimension. We have 16 learnable queries with six encoder-decoder layers of transformer attention, each layer consists of 8 heads. We then compute multi-head cross-attention between each query vector and the encoded 3d points. This gives us a map of size $M \times 512 \times 16$, where $M$ is the number of heads in the multi-head attention.

Each attention map is individually upsampled through a PointTransformer decoder of the architecture 512-16-64, 1024-16-64, 1024-16-64, which gives us the final instance segmentation mask per query, similar to DETR (Carion et al., 2020). We then use Hungarian matching to match the predicted masks against the ground truth masks and then apply binary cross entropy loss for each match. We aggregate the losses from each query and backpropagate. Figure 10 visualizes the architecture of 3D-DETR. Note that we do not follow the 2 stage training of Carion et al. (2020), rather we train their model end-to-end for instance segmentation, similar to our model. We have found this trick to save compute and still not harm the end results.

**Learning2Group.** (Luo et al., 2020)   Learning2Group progressively groups points into segments by learning pairwise grouping decisions parameterized by features of the point clusters to be grouped. Given the intermediate grouping decisions are not supervised, and the non-differentiability of their grouping functions they use reinforcement learning gradients for training using supervision from the final segmentation. We use their open-sourced architecture and code for comparision with our mode. We train their model using our datasets from scratch.

**Shape2Prog** (Tian et al., 2019)   Shape2Prog is a shape program synthesis method that is trained supervised to predict shape programs from object 3D point clouds. Shape2Prog introduced two synthetically generated datasets that helped the model parse 3D pointclouds from ShapeNet Chang et al. (2015) into shape programs without any supervision: i) Generic primitive set (Figure 8) we discussed earlier in which they use to pre-train their part decoders, and ii) Synthetic whole shape dataset of chairs and tables (Figure 9) generated programmatically alongside its respective ground-truth programs. Their model requires supervised pre-training on the dataset of synthetic whole shapes paired with programs. We therefore use their publicly available model weights trained on synthetic whole shapes to further train on PartNet Chairs. Note that no other baseline nor Slot-TTA assumes access to the synthetic whole shape dataset. We use their open-sourced architecture, code and pretrained checkpoints for comparision with our model. We change the value of number of blocks similar to the number of slots in our model for each dataset.

**PQ-Nets.** (Wu et al., 2020)  PQ-Nets is a sequential encoder-decoder architecture, that takes 3D point cloud as input and sequentially encodes it into multiple 1D latents which are then decoded to part point clouds. It achieves this decomposition by pre-training their decoder to predict part point clouds. We use their open-sourced architecture and code for comparision with our model. We train their model using our datasets from scratch. We change the value of number of slots in their model based on the maximum number of parts in the dataset.

## 9  ADDITIONAL EXPERIMENTS

### 9.1  SEGMENTING RGB IMAGES IN MULTI-VIEW SCENES

| Method | in-dist (5-7 instances) | | out-of-dist (16-30 instances) | |
|---|---|---|---|---|
| | before TTA | after TTA | before TTA | after TTA |
| Slot-TTA-SlotMixer_Decoder | **0.94** | 0.89 | 0.65 | 0.72 |
| Slot-TTA-SRT_Decoder | 0.92 | 0.88 | 0.60 | 0.63 |
| Slot-TTA-tta_All_param | N/A | 0.92 | N/A | 0.82 |
| Slot-TTA-tta_Norm_param | N/A | 0.94 | N/A | 0.79 |
| Slot-TTA-tta_Slot_param | N/A | 0.94 | N/A | 0.76 |
| Slot-TTA w/o Weighted_Sample | N/A | 0.93 | N/A | 0.81 |
| Slot-TTA (Ours) | 0.92 | **0.95** | **0.70** | **0.83** |

Table 5: **ARI Segmentation accuracy (higher is better)** in the in-distribution test set of 5-7 object instances and out-of-distribution 16-30 object instances.

| Method | in-dist (ShapeNet categories) | | out-of-dist (GSO categories) | |
|---|---|---|---|---|
| | before TTA | after TTA | before TTA | after TTA |
| Mask2Former | 0.93 | N/A | **0.93** | N/A |
| Mask2Former+BYOL | 0.93 | 0.95 | 0.92 | 0.93 |
| Mask2Former+Recon | 0.93 | 0.92 | 0.92 | 0.91 |
| Slot-TTA (Ours) | 0.92 | **0.95** | 0.92 | **0.95** |

Table 6: **ARI Segmentation accuracy (higher is better)** in the in-distribution test set of ShapeNet object categoriesChang et al. (2015) and out-of-distribution test set of GSO object categories Downs et al. (2022).

In Table 6, we tested our model on a different distribution shift. In the test set instead of increasing the number of instances in the scene in Table 1, we introduced instances from new object categories. Specifically the MSN Sajjadi et al. (2022b) train-set consists of ShapeNet object categoriesChang et al. (2015) (Tables, Chairs etc), whereas the new test-set consists of Google Scanned Object Downs et al. (2022) (GSO) categories (Shoes, Stuffed toys etc). We find that our model gets a score of 0.92 before TTA and 0.95 after TTA, whereas mask2former and its TTA counterparts get a score of 0.93, clearly demonstrating the benefit of slot-centric test-time adaptation over a state-of-the-art baseline.

We conduct various ablations of Slot-TTA in Table 1. In Figure 11, we show additional qualitative results comparing Slot-TTA-Fast and Slot-TTA-Slow.

(i) We ablate different decoder choices in the topmost section where instead of using the broadcast decoder we use the Scene representation transformer (SRT) decoder (Sajjadi et al., 2022b) which we refer to as **Slot-TTA-SRT_Decoder**  or the SlotMixer decoder (Sajjadi et al., 2022a), referred to as **Slot-TTA-SlotMixer_Decoder**.

(ii) We ablate what parameters to adapt at test time. As it's unclear since TENT  (Wang et al., 2020) optimizes BatchNorm or LayerNorm parameters, but TTT  (Sun et al., 2020) optimizes the shared parameters between the SSL and the task-specific branch, which in our case will be all the parameters in the network. In Table 5, **Slot-TTA-tta_All_param** is when we adapt all the

network parameters, **Slot-TTA-tta_Norm_param** adapts only the Layer or BatchNorm parameters and **Slot-TTA-tta_Slot_param** adapts only the learnable slot embeddings. We find that optimizing only the encoder parameters works the best for our setting.

(iii) Further, we ablate error-conditioned pixel sampling where **Slot-TTA w/o Weighted_Sample** refers to our model that uses uniform sampling instead of the error weighted sampling.

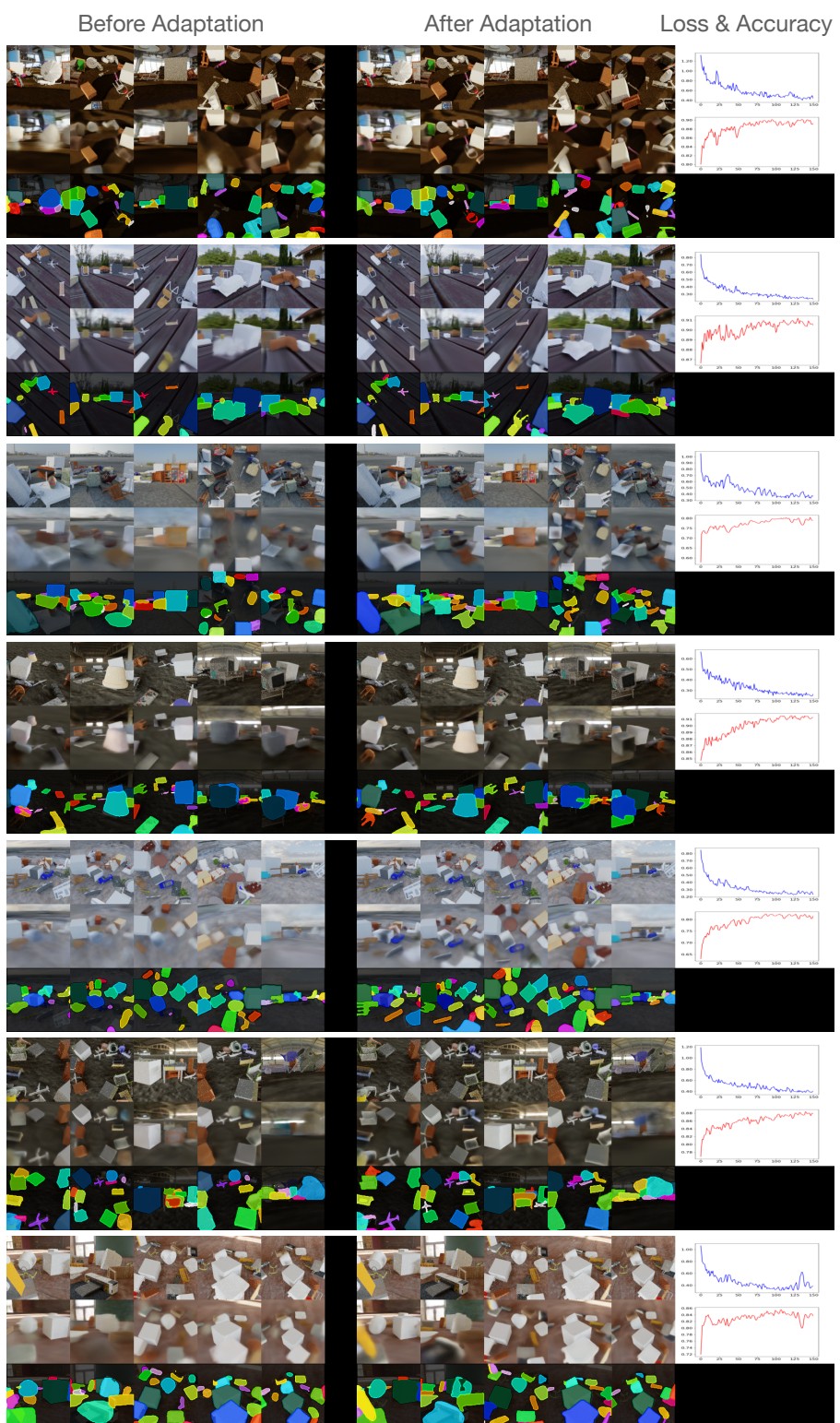

Figure 11: On the left, we visualize Slot-TTA-Fast. In the middle, we visualize Slot-TTA-Slow. In the first row we visualize the ground truth target RGB views. In the second and third row we visualize Slot-TTA predicted target RGB views and their segmentation masks. On the right-most column we visualize the RGB loss and segmentation accuracy when doing slow inference. Same setting as Section 4.1

## 9.2 SEGMENTING SINGLE-VIEW RGB IMAGES

In Figure 12 we qualitatively compare Slot-TTA-Slow with Slot-TTA-Fast. We show that slow inference can help discover objects missed by Slot-TTA. We also show some failure cases where slow inference could override the object-centric bottleneck to achieve higher reconstruction accuracy.

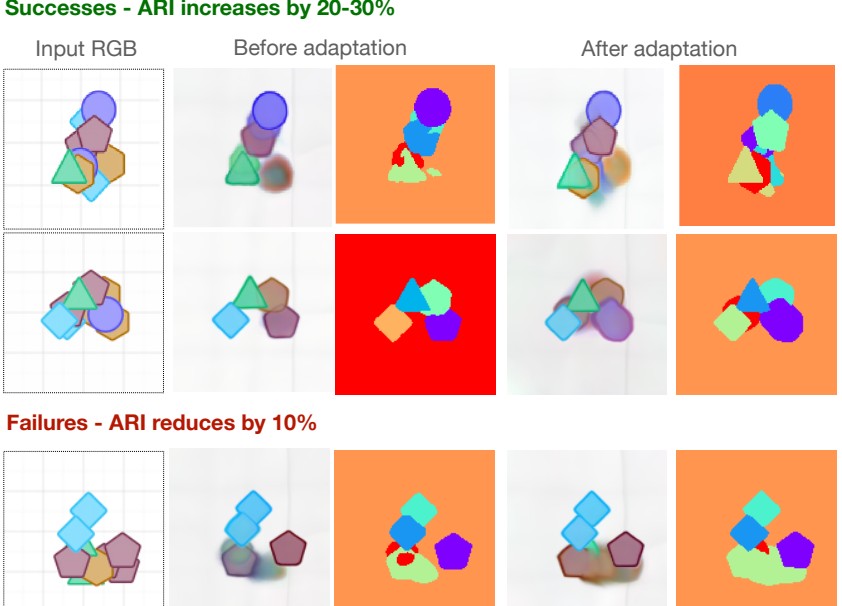

Figure 12: Success and Failure cases of slow-inference on Multi-Shape dataset. Same setting as Section 4.2

## 9.3 SEGMENTING 3D POINT CLOUDS

### 9.3.1 SUPERVISION FROM A DATASET OF GENERIC 3D PART PRIMITIVES

In this Section we show additional qualitative and quantitative results for Section 4.3.1. Neither Slot-TTA nor the baselines have access to ground-truth 3D segmentations during training of Chair or Table category; as a result, they may output 3D parts of coarser or finer resolution. PartNet contains three different levels of ground-truth segmentation labels with progressively finer segmentation granularity. In Table 7, we further extend Table 3 to include different levels in Chair and Table category. Here $(X/Y/Z)$ refers to (level 1/level 2/level 3) scores respectively. We pick the best performing level in fast inference and specifically report it's slow inference results. We further qualitatively compare our model against Shape2Prog (best performing baseline) in Figure 13

| Method | in-dist (Chairs) | | out-of-dist (Tables) | |
|---|---|---|---|---|
| | Fast Infer. | Slow Infer. | Fast Infer. | Slow Infer. |
| Shape2Prog Tian et al. (2019) | **0.28**/0.21/0.26 | 0.53 | 0.21/0.23/**0.23** | 0.40 |
| PQ-Nets Wu et al. (2020) | **0.20**/0.18/0.16 | 0.31 | **0.17**/0.14/0.16 | 0.21 |
| Slot-TTA | 0.51/0.48/**0.57** | **0.62** | 0.51/0.55/**0.60** | **0.69** |

Table 7: **ARI Segmentation accuracy (higher is better)** in the test set of Chair (in-distribution) and Table category of PartNet(out-of-distribution). Slot-TTA significantly outperform all of the baselines.

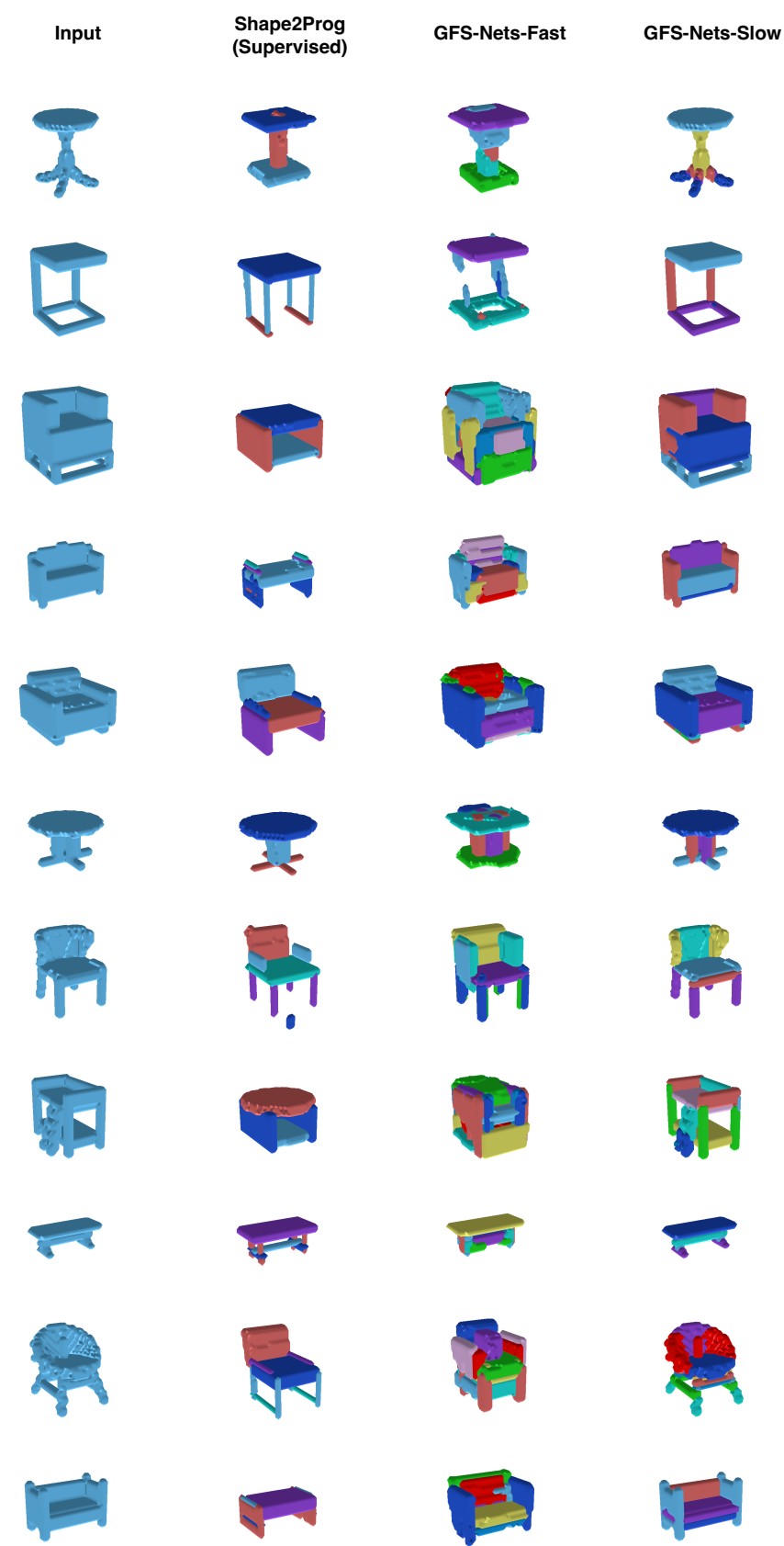

Figure 13: Additional segmentation results on out-of-distribution categories when supervised from generic primitives. Same setting as Section 4.3.1

### 9.3.2 SUPERVISION FROM A RELATED OBJECT CATEGORY

In this section we further qualitatively and quantitatively compare our model against baselines on the task of instance segmentation. We follow the same experimental setup of Section 4.3.2, where we use Chair category segmentation supervision and test for cross-category generalization. We test the segmentation accuracy of the models on other categories of PartNet in Table 9. Note that **a few of these categories don't share any common parts** with the Chair category that's why the absolute scores of Slot-TTA and all other baselines are very low on these categories. We find this specific metric to be not well-defined, as generalizing from Chairs to Shoes or Vase, might not make sense, but to a furniture category like a Table or Bed does. Inspite of this overall, we find the mean score of Slot-TTA to outperform all other baselines.

Further in Table 8, we tested our model on an object detection task, where we predict the bounding box for each instance. We find that our model gets a similar performance improvement as our instance segmentation task after test-time adaptation. Specifically the box mIoU improves from 0.51 to 0.63 after TTA, whereas our 3D-DETR baseline gets a score of 0.53. Thus showcasing that our method generalizes beyond the task of instance segmentation.

| Method | in-dist (Chair) | | out-dist (Table) | |
|---|---|---|---|---|
| | Fast Infer. | Slow Infer. | Fast Infer. | Slow Infer. |
| 3D-DETR (Misra et al., 2021) | **0.68** | N/A | **0.53** | N/A |
| Slot-TTA | 0.62 | **0.65** | 0.51 | **0.63** |

Table 8: **Object detection box mIoU accuracy** (higher is better) in the test set of Chair category (in-distribution) and Table category (out-of-distribution) when trained using the supervision of Chair category.

| Method | Display | Bottle | Clock | Dishwasher | Door | Earphone | Faucet |
|---|---|---|---|---|---|---|---|
| 3D-DETR | 0.21 | 0.25 | 0.12 | 0.08 | **0.19** | 0.21 | 0.28 |
| Learning2Group | 0.30 | 0.19 | **0.18** | **0.10** | 0.17 | 0.25 | 0.28 |
| Slot-TTA-Fast | 0.16 | 0.26 | 0.04 | 0.04 | 0.09 | 0.18 | 0.25 |
| Slot-TTA-Slow | **0.53** | **0.45** | 0.05 | 0.06 | 0.12 | **0.32** | **0.37** |

| Method | Knife | Lamp | Microwave | Refrigerator | TrashCan | Vase | Bed |
|---|---|---|---|---|---|---|---|
| 3D-DETR | 0.23 | 0.21 | 0.01 | **0.17** | 0.17 | 0.14 | 0.26 |
| Learning2Group | 0.28 | 0.17 | **0.14** | 0.1 | 0.15 | **0.25** | 0.32 |
| Slot-TTA-Fast | 0.24 | 0.10 | 0.01 | 0.04 | 0.11 | 0.10 | 0.28 |
| Slot-TTA-Slow | **0.38** | **0.24** | 0.01 | 0.10 | **0.30** | 0.15 | **0.41** |

| Method | Mean | | | | | | |
|---|---|---|---|---|---|---|---|
| 3D-DETR | 0.20 | | | | | | |
| Learning2Group | 0.22 | | | | | | |
| Slot-TTA-Fast | 0.15 | | | | | | |
| Slot-TTA-Slow | **0.27** | | | | | | |

Table 9: **ARI Segmentation scores (higher is better)** on the test set of the other level-3 categories(out-of-distribution) of PartNet. All the models are trained using the training set of the Chair category in PartNet.

**3D-DETR**    **GFS-Nets-Slow**    **3D-DETR**    **GFS-Nets-Slow**

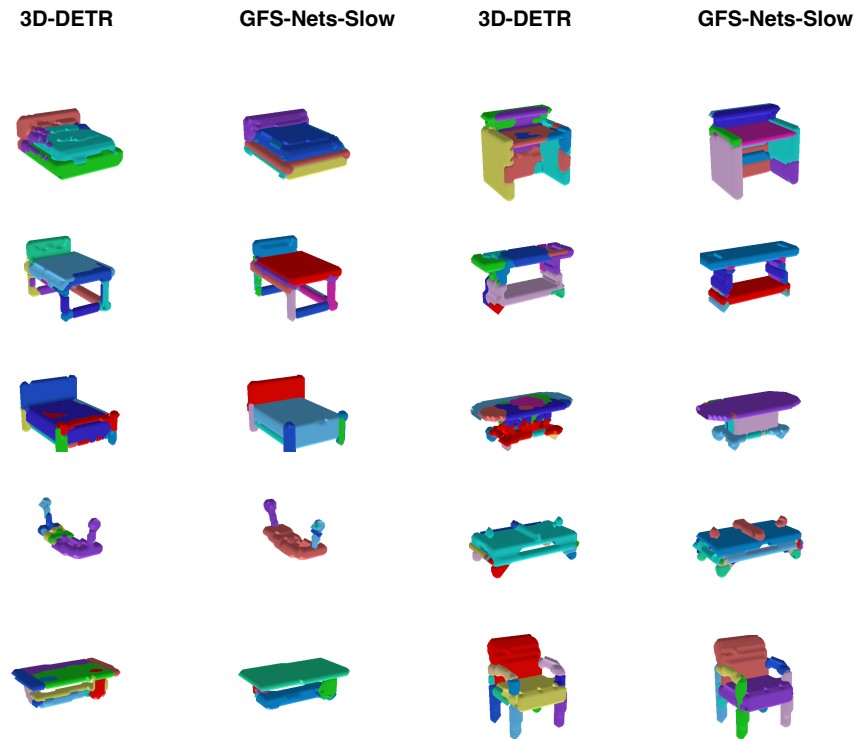

Figure 14: Additional segmentation results in out-of-distribution categories when supervised from a related object category. Same setting as Section 4.3.2

## 10 ADDITIONAL RELATED WORK

**Shape program synthesis and analysis-by-synthesis**    Slot-TTA is also related to works in analysis-by-synthesis Kulkarni et al. (2015), program synthesis for shape prediction Tian et al. (2019); Ellis et al. (2020); Li et al. (2020), as well as earlier works on Computer Vision, such as Marr's 3D sketch Marr (1982) which involves representing a scene in terms of generalized cylinders and their syntactic relations to each other. In place of data-driven Markov Chain Monte Carlo search of analysis-by-synthesis methods that require good initialization, our slow inference searches in the space of primitives by gradient descent. In contrast to program synthesis methods, it does not require a predefined domain-specific language (DSL) or program annotations for visual structures Li et al. (2020), rather, it discovers compositions over primitives via its slow inference.

**Unsupervised 3D Part Discovery**    There are numerous methods that attempt the decomposition of complex 3D shapes into primitive parts without primitive supervisionKato & Harada (2019); Genova et al. (2019); Paschalidou et al. (2020); Gao et al. (2019); Deprelle et al. (2019); Tulsiani et al. (2017); Deng et al. (2020); Chen et al. (2019). Traditional primitives include cuboids Tulsiani et al. (2017); Niu et al. (2018), superquadrics Paschalidou et al. (2019; 2020), and convexes Deng et al. (2020); Chen et al. (2020). Genova et al. (2020) proposes a 3D representation that decomposes space into a structured set of implicit functions Genova et al. (2019). Neural Parts Paschalidou et al. (2021) represents arbitrarily complex genus-zero shapes and thus yields comparatively expressive parts. However the resulting parts of these methodsGenova et al. (2020); Paschalidou et al. (2021) are still not semantically meaningful and the decomposition is highly dependent on the number of parts initialized. The work of Yao et al. (2021) does 3D part reconstruction directly from a 2D image input without access to any ground-truth 3D shapes for training. However, both Yao et al. (2021) and Paschalidou et al. (2021) take as input the number of parts, and different decompositions are predicted with varying part numbers. There is no clear way to select the right number of parts. In our case, parts can be quite complex: pairs of parallel surfaces, quadruplets of legs, as we use implicit

functions to represent them. Moreover Slot-TTA's dynamic attention-based routing allows it to infer different number of parts for each input scene.

