# OpenReview forum: "Test-time Adaptation for Segmentation via Image Synthesis"
_ICLR.cc/2023/Conference — Submitted to ICLR 2023_

### Official Review · Reviewer_f456 · 2022-10-20

**Confidence:** 4
**Correctness:** 3
**Technical Novelty And Significance:** 1
**Empirical Novelty And Significance:** 1
**Recommendation:** 3

**Clarity, Quality, Novelty And Reproducibility:**

Overall, the contribution of this paper is too incremental, and the comparison to the state-of-the-art methods is missing. The overall quality of this paper is too weak to be accepted to ICLR.

**Strength And Weaknesses:**

1. Strength
- The proposed test time domain adaptation method outperforms the normal method, which is never adapted at test time.
- This paper compared not only the instance segmentation but also the 3D segmentation.

2. Weakness
- This paper combines two existing methods slot attention and the object scene representation transformer. Instance query-based matching is a common pipeline for semantic segmentation.
- The contributions to the test time adaptation method are marginal. The only loss that is applied for adaptation is the reconstruction loss in eq 3. Reconstruction loss is common for self-supervised loss. Applying only reconstruction loss can degrade the performance of the other test datasets.
- The proposed method is not densely evaluated. Only a small number of classes and datasets are utilized for the evaluation.
- The proposed method has not been compared to the other test time domain adaptation methods [1-3].  If the self-supervised loss is effective, why not apply the loss to the other tasks such as semantic segmentation, and object detection? The proposed method only compared with the normal algorithm, which is not applied test time domain adaptation, such as mask2former.


[1] Dequan Wang, Evan Shelhamer, Shaoteng Liu, Bruno Olshausen, and Trevor Darrell. Tent: Fully test-time adaptation by entropy minimization. In ICLR, 2021.
[2] Yu Sun, Xiaolong Wang, Zhuang Liu, John Miller, Alexei Efros, and Moritz Hardt. Test-time training with self-supervision for generalization under distribution shifts. In International conference on machine learning, pp. 9229–9248. PMLR, 2020.
[3] Inkyu Shin, Yi-Hsuan Tsai, Bingbing Zhuang, Samuel Schulter, Buyu Liu, Sparsh Garg, In So Kweon, and Kuk-Jin Yoon. Mm-tta: Multi-modal test-time adaptation for 3d semantic segmentation. In CVPR, 2022.

**Summary Of The Paper:**

This paper proposes a domain adaptation method in test time. To do so, this paper proposes a semi-supervised slot-centric approach that combines slot attention and object scene representation transformer. The model adapts to a single test sample without supervision at test time. It is optimized for the self-supervised objective function. This paper shows that the loss for image synthesis is helpful for the test-time domain adaptation.


**Summary Of The Review:**

Because of (1) Incremental technical contribution and (2) Lack of comparison to the existing method, this paper is not adequate to be accepted to ICLR.

---

> ### Author Response · Authors · 2022-11-09
> **Response to review part(1/2)**
>
> We thank the reviewer for their feedback. Below we address your concerns.
>
> **Q1. Very incremental work.**
>
> As Reviewer 2 points out, "this is the first work exploring the scene understanding tasks by test-time adaptation". Given that scene understanding is a central task in computer vision and given there is extensive literature on TTA for the task of image classification, we believe TTA for scene understanding is an important research direction, and our paper makes a first and a significant step in that direction.
>
> The TTA community largely focuses on different types of SSL losses, such that the gradients of the SSL loss align with the gradients from the task loss (eg. Sun et al., 2020; Bartler et al., 2022; Gandelsman et al., 2022). However, we find in our work (Table 1), specifically in the Mask2former+Recon and Mask2former+BYOL baselines, that searching only amongst different loss functions could be insufficient. It is important to look at architectural biases, specifically slot-centric biases to obtain additional improvement during TTA, especially for the task of scene understanding.
>
> Further, slot-based generative models (eg. Engelcke et al., 2019; Greff et al., 2019; Locatello et al., 2020) have not been used or developed with the forethought of test time adaptation. As a matter of fact, Engelcke et al (2020), show that TTA through reconstruction in slot-centric generative models fails, due to a reconstruction-segmentation trade-off: as the entity bottleneck gets wider, reconstruction improves but then the segmentation gets worse. In this work, we show that weak supervision can help break this trade-off and test-time adaptation using reconstruction objectives can significantly help scene decomposition.
>
> Additionally, simply mixing slot-centric architectures with TTA is not sufficient. There are additional design choices that come into play. For example, in OSRT (Sajjadi et al., 2022a) it has been found SlotMixer decoder and Broadcast decoder do equally well in unsupervised object discovery, however in our experiments (Table 5) for TTA we find the Broadcast Decoder to significantly outperform (by 11 points) the SlotMixer decoder in TTA (0.72 -> 0.83). Further finding the right set of parameters to adapt, pixel-sampling strategy, and how to supervise slot-centric models are important design choices and play an important role in this improvement (Table 5).
>
> We have provided the above-mentioned insights in Section 4.3.1 results subsection and in Introduction paragraphs 2,3 and 4.
>
> **Q2. Reconstruction loss can degrade performance over other test datasets.**
>
> Indeed, when performing TTA it is generally possible that performance degrades on the other datasets. However, in our setting we conduct TTA on a single example without modifying the original weights permanently. In this way we can perform TTA for each sample **independently**, while additionally performance on the original test sets will not be affected. Furthermore, we note that per-example TTA is not computationally heavy as it takes less than 3 seconds (less than 50 iterations) for getting 90% of the boost in performance.
>
> **Q3. Proposed method is not densely evaluated.**
>
> >“Only a small number of classes”
>
> We test our method using **all the 15 categories** in PartNet in our Supplementary section 9.3.2, as pointed out in our paper Section 4.3.2. PartNet is a popular benchmark for part segmentation on point clouds, and it has been used as a primary dataset for evaluating different part segmentation models (Luo et al., 2020, Tian et al., 2019, Wu et al., 2020).
>
> >“Only a small number of datasets”
>
> We test our model using 3 datasets each in a different modality, which goes far beyond the diversity of datasets considered in prior work on TTA. We further tested our model with 3 additional datasets, CLEVR [1] (Section 4.1), RoomDiverse [2] (Section 4.1) and Plating (Section 4.2), we found similar performance improvements as reported in our paper.  On CLEVR, with slow inference, our model improves performance from **0.91 to 0.97**, where in the train set we use 4-7 objects and in the test set we use 7-10 objects.  On RoomDiverse, our scores improve from **0.74 to 0.81** on a similar train-test split as above.  For further real-world results, we run our model on a real-world salad plating dataset that we collected, and we will publicly release it. Results: https://sites.google.com/view/slottta/home. As you can see, our model effectively segments the scene into objects and amodally reconstructs each one, despite heavy occlusions.
>
> **References:**
>
> [1] Johnson, Justin, et al. "Clevr: A diagnostic dataset for compositional language and elementary visual reasoning." Proceedings of the IEEE conference on computer vision and pattern recognition. 2017.\
>
> [2] Yu, Hong-Xing, Leonidas J. Guibas, and Jiajun Wu. "Unsupervised discovery of object radiance fields." arXiv preprint arXiv:2107.07905 (2021).

---

> > ### Author Response · Authors · 2022-11-09
> > **Response to review part(2/2)**
> >
> > **Q4. Proposed method not compared to other TTA method**
> >
> > >“The proposed method only compared with the normal algorithm, which is not applied test time domain adaptation, such as mask2former.”
> >
> > This is incorrect.  We compare our method against MT3 (Bartler et al., 2022) which is the current SOTA on classification with test-time adaptation (Mask2former-BYOL). MT3 beats [2], thus we don't compare against [2]. Additionally, we compare our method to Mask2former-RECON, which does TTA with a reconstruction objective similar to our model.
> >
> > >"The proposed method has not been compared to the other test time domain adaptation methods [1-3]."
> >
> > [1] and [3] are pseudo-labeling approaches, thus requiring more than one example for test-time adaptation; they need a dataset!  They specifically need confident and accurate predictions that they can propagate to other examples.
> >
> > Our method on the other hand can adapt to a **single example at test-time**, as we use an SSL loss for adaptation.
> >
> > Furthermore, we find [1] and [3], to be orthogonal approaches that are not competing with our method. As when given many examples, they can easily be incorporated into any method as they are task and architecture agnostic.
> >
> > **Q5. Comparisons to semantic segmentation or object detection tasks?**
> >
> > We chose instance segmentation as it’s a much more complex and fine-grained task than semantic segmentation or object detection [4]. For example object detection provides a bounding boxes for each instance, where as semantic segmentation labels each
> > pixel according to the object class. Instance segmentation on the other hand, does both by labeling each pixel based on instance grouping.
> >
> > Further, it is straightforward to retrieve bounding boxes or semantic segments given accurate instance segments. For example one could simply find the farthest corners of the predicted mask to get the bounding boxes or they could merge instance masks belonging to the same object classes to get semantic segments.
> >
> > In the meantime during the rebuttal process, we will try our best to get scores for the object detection task mentioned above by the reviewer.
> >
> >
> > **References:**
> >
> > [4] Hafiz, Abdul Mueed, and Ghulam Mohiuddin Bhat. "A survey on instance segmentation: state of the art." International journal of multimedia information retrieval 9.3 (2020):

---

> > > ### Author Response · Authors · 2022-11-15
> > > **Results on object detection and comparison against pseudo-labelling methods**
> > >
> > > **Results on object detection**
> > >
> > > To satisfy your concern we tested our model on an object detection task in section 4.3.2, where we predict the bounding box for each instance along with its instance segmentation mask, and we supervise both heads. We find that our model gets a similar performance improvement as our instance segmentation task after test-time adaptation. Specifically the box mIoU improves from **0.51** to **0.63** after TTA, whereas our 3D-DETR baseline gets a score of **0.53**.
> > >
> > > **Comparison to pseudo-labelling methods**
> > >
> > > To satisfy your request, we compared our method against TENT (Wang et al., 2020), where instead of optimizing per example over reconstruction loss (like in our model and Mask2former+Recon baseline), we optimized its entropy over each pixel. We found after TTA  both our model and the baselines ARI score to **reduce by 2 to 3 points** respectively.  We tried different tricks such as optimizing only the normalization parameters, but found a similar reduction in performance. We think this is because pseudo-labelling is primarily to be used with multiple examples (i.e. a dataset) instead of a single example.

---

> > ### Author Response · Authors · 2022-11-18
> > **Request for final feedback.**
> >
> > Dear Reviewer,
> >
> > We hope we have been able to address all of your concerns in the updated manuscript.
> >
> > As the rebuttal process ends tomorrow, please let us know if you have any further comments or queries.
> >
> > Thank you.

---

### Official Review · Reviewer_MJUr · 2022-10-24

**Confidence:** 2
**Correctness:** 3
**Technical Novelty And Significance:** 3
**Empirical Novelty And Significance:** 3
**Recommendation:** 6

**Clarity, Quality, Novelty And Reproducibility:**

The paper is clearly written and the quality is good.

A novel idea of test-time adaptation is introduced in this paper for segmentation

The code and datasets are not shared.

**Strength And Weaknesses:**

The paper has the following strenghts:

1) The paper is well-written the main ideas are effectively presented.
2) The paper tackles a problem of interest to the community where out of distribution segmentation task can be handled.
3) The slow-inference results are better on the benchmark datasets.


The weakness of the paper in my opinion are:
1) Some of hte limitation of the methods are discussed in the paper and the one which I think is most signifcant is the absence of the heirarchy of components of object. In segmentation the heirarcy and relations of components of objects is really important.

2) The fast infrence results are slightly lower than the benchmarks.



**Summary Of The Paper:**

The paper proposes Generating Fast and Slow Networks (GFS-Nets), where the model has a slot-centric image or point-cloud rendering part, which is adpated per scene at test time through gradient decent. This adaptation at test time improves performance on out of distribution data.



**Summary Of The Review:**

In my opinion the paper is well written and tackles a problem of interest to the community. I have, however, outlined some of the aspects which i think need some explanation in the strengths and weakness section.

---

> ### Author Response · Authors · 2022-11-09
> **Response to review**
>
> We are encouraged that you found our paper to have a "novel idea", have "good results", "tackle an important problem" and be "well-written". Below we address your concerns.
>
> **Q1. Absence of part-whole hierarchy.**
>
> Yes, we agree with the reviewer that finding hierarchical slots is important. As this allows us to achieve combinatorial generalization across new object categories sharing similar parts and part-structure. Currently this is an open research area and there are no or few slot-based backbones capable of this.
>
> However the insights of our work that architectural inductive biases are important  for instance segmentation TTA are in principle agnostic to whether the backbone yields part-object or just object decomposition.
>
> **Q2. Fast-inference scores are lower.**
>
> Indeed, the fast inference scores are lower than the Mask2former baseline. We think this is due to our slot bottleneck, as it limits the information flow, thus hindering the network’s expressivity and capability of fitting on the training set. To test this hypothesis instead of predicting the masks from the slot-bottleneck, we made the slots to directly attend to the grid features (similar to Mask2former) thus removing the information bottleneck. With this new change we find the fast-inference scores to match with Mask2former; however then the slow-inference scores don’t improve.
>
> To address this problem, we think a dynamic bottleneck can help; for example, we loosen the information bottleneck during training on the train set but have tighter bottlenecks during TTA. We leave this for future work.

---

> ### Author Response · Authors · 2022-11-18
> **Request for final feedback.**
>
> Dear Reviewer,
>
> We hope we have been able to address all of your concerns in the updated manuscript.
>
> As the rebuttal process ends tomorrow, please let us know if you have any further comments or queries.
>
> Thank you.

---

### Official Review · Reviewer_Hiry · 2022-10-25

**Confidence:** 1
**Clarity, Quality, Novelty And Reproducibility:** NA
**Correctness:** 3
**Technical Novelty And Significance:** 3
**Empirical Novelty And Significance:** 3
**Recommendation:** 6

**Strength And Weaknesses:**

NA

**Summary Of The Paper:**

NA

**Summary Of The Review:**

NA

---

### Official Review · Reviewer_Ad2e · 2022-10-31

**Confidence:** 4
**Correctness:** 3
**Technical Novelty And Significance:** 2
**Empirical Novelty And Significance:** 2
**Recommendation:** 3

**Clarity, Quality, Novelty And Reproducibility:**

The clarity and quality of this work, from my point of view, are not enough for ICLR. The novelty and technical contribution are limited. I believe this work is reproducible.

**Strength And Weaknesses:**

Strengths:

-- As far as I know, this is the first work exploring the scene understanding tasks by test-time adaptation.

-- The authors conduct extensive experiments on three different tasks with different kinds of input, i.e., posed multi-view 2D RGB image, single-view 2D RGB images, 3D point clouds. Correspondingly, the authors conduct modifications on the introduced network based on different inputs.

Weakness:

-- The technical contribution and novelty of this work seem not strong enough for ICLR. For example, the basic modules, and encoder/decoder structure of GFS-Nets, are all based on prior works (OSRT, spatial broadcast decoder, etc). It seems there is few new designs in the network structure.

-- For the test-time adaption part, it seems that the authors just directly borrow the idea and apply it to their tasks. Without a technical contribution, it is hard to claim there is a significant contribution in this part.

-- The submission is not self-contained and some formulations/terms need refined. For example, what is Hungarian matching? What is the occupancy value O_i? The authors should make the content self-contained as not all authors have related backgrounds. In Eq. 3, the formulation of each loss term is not provided in this work. Do the three tasks in this work use the same formulation as the l_{seg}? It should be {c_k, a_k} = Dec(p, S_k).

**Summary Of The Paper:**

This work targets to segmenting scenes into objects and parts under an out-of-distribution setting. In an out-of-distribution setting, the statistical distribution of the testing example is different from that of the training examples. To handle the problem, the authors build a Generating Fast and Slow Network (GFS-Net) based on prior works, such as slot-centric generative models. The GFS-Net combined with test time adaption strategy, improves the segmentation performance on three different tasks. The authors conduct extensive experiments to demonstrate the efficacy of their solution.

**Summary Of The Review:**

In summary, this work is interesting and practical. However, the technical contribution is limited, making the novelty not strong enough. The writing and organization of this paper need significant improvement.

---

> ### Author Response · Authors · 2022-11-09
> **Response to review**
>
> We are encouraged that you found our work to have "extensive experimentation" . Below we address your concerns.
>
> **Q1. Lack of novelty.**
> > “the basic modules, and encoder/decoder structure of GFS-Nets, are all based on prior works “
>
> We agree with the reviewer that the *architectural* novelty of the proposed method is limited. However the idea of combining slot-centric models with TTA is novel and leads to a number of significant insights and avenues for future work in TTA and slot-centric models.
>
> >“For the test-time adaption part, it seems that the authors just directly borrow the idea and apply it to their tasks”
>
> As you pointed out, ”this is the first work exploring the scene understanding tasks by test-time adaptation”. Given that scene understanding is a central task in computer vision and given there is extensive literature on TTA for the task of image classification, we believe TTA for scene understanding is an important research direction, and our paper makes a first and a significant step in that direction.
>
> The TTA community largely focuses on different types of SSL losses, such that the gradients of the SSL loss align with the gradients from the task loss (eg. Sun et al., 2020; Bartler et al., 2022; Gandelsman et al., 2022). However, we find in our work (Table 1), specifically in the Mask2former+Recon and Mask2former+BYOL baselines, that searching only amongst different loss functions could be insufficient. It is important to look at architectural biases, specifically slot-centric biases to obtain additional improvement during TTA, especially for the task of scene understanding.
>
> Further, slot-based generative models (eg. Engelcke et al., 2019; Greff et al., 2019; Locatello et al., 2020) have not been used or developed with the forethought of test time adaptation. As a matter of fact, Engelcke et al (2020), show that TTA through reconstruction in slot-centric generative models fails, due to a reconstruction-segmentation trade-off: as the entity bottleneck gets wider, reconstruction improves but then the segmentation gets worse. In this work, we show that weak supervision can help break this trade-off and test-time adaptation using reconstruction objectives can significantly help scene decomposition.
>
> Additionally, simply mixing slot-centric architectures with TTA is not sufficient. There are additional design choices that come into play. For example, in OSRT (Sajjadi et al., 2022a) it has been found SlotMixer decoder and Broadcast decoder do equally well in unsupervised object discovery, however in our experiments (Table 5) for TTA we find the Broadcast Decoder to significantly outperform (by 11 points) the SlotMixer decoder in TTA (0.72 -> 0.83). Further finding the right set of parameters to adapt, pixel-sampling strategy, and how to supervise slot-centric models are important design choices and play an important role in this improvement (Table 5).
>
> We have provided the above-mentioned insights in Section 4.3.1 results subsection and in Introduction paragraphs 2,3 and 4.
>
> **Q2. Re-formulation of terms.**
>
> We thank the reviewer for pointing out these issues, we will add the below explanations to the paper.
>
>
> >“What is Hungarian matching?”
>
> Hungarian matching is a combinatorial optimization algorithm that solves assignment problems in polynomial time. For example, given a MxN cost matrix of assigning M predicted segments with N ground truth segments, it finds an assignment for each predicted segment with the ground truth segment, such that each predicted segment is assigned to a single ground truth segment while minimizing the total cost of assignment.
>
> >“What is occupancy value O_i”
>
> $O_i$ (occupancy value) represents the predicted probability of the queried point location being occupied or present in the 3D volumetric space. Specifically it is  $o_{i,x,y,z} = Dec(S_i ,(x, y, z))$ as mentioned in Section 3.2.1.
>
>
>
> > “Do the three tasks in this work use the same formulation as the l_{seg}? “
>
> Yes, all three tasks use categorical cross entropy loss for supervision, represented by $l_{seg}$. However for RGB images we supervise the alpha masks that are $a_k$, but for point clouds we directly supervise $o_k$, since there is no color term $c_k$  for point clouds. We mention this in section 3.2.2 and more details can be found in the point cloud section of 8.3 Section in supplementary.

---

> ### Author Response · Authors · 2022-11-18
> **Request for final feedback.**
>
> Dear Reviewer,
>
> We hope we have been able to address all of your concerns in the updated manuscript.
>
> As the rebuttal process ends tomorrow, please let us know if you have any further comments or queries.
>
> Thank you.

---

### Official Review · Reviewer_VgmG · 2022-11-02

**Confidence:** 3
**Correctness:** 4
**Technical Novelty And Significance:** 2
**Empirical Novelty And Significance:** 3
**Recommendation:** 5

**Clarity, Quality, Novelty And Reproducibility:**

- Clarity&quality: it's easy to follow overall. The method section can be better with figures for the model architecture for three different settings.

- Novelty: It's incremental without much new insights.

- Reproducibility: good.

**Strength And Weaknesses:**

Strength
- It's an interesting idea to take the slot-centric approach to do TTA with the image reconstruction auxiliary task, which is not done before.

- Strong performance on popular datasets from two modalities.

- Detailed ablation studies and model description for design justification and reproducibility.

Weakness
- Lack of novelty. The proposed method combines two existing works: slot-attention-based instance segmentation model and the TTA method with image reconstruction auxiliary loss, without providing much new insights.

- Biased experiment setting. For image instance segmentation, the benchmark dataset (sec. 7.1-7.2) has few instances in the training set and much more instances in the test set. Such experiment design favors instance segmentation models for crowded scenes (i.e., slot-attention based method) over popular maskformer models. Thus, it's unclear if the performance gain over the baseline (maskformer+reconstruction) majorly comes from the inductive bias of the backbone selection instead of the TTA framework. Also, It's unclear if the proposed method is indeed better on common images (e.g., MS-COCO). The paper result can be more convincing if it shows comparison on natural images.

- Naming. (1) Task: the proposed method is specifically designed for the "instance segmentation" task, but the paper uses "segmentation" in many places, which can be confusing. (2) Title: It's more accurate to use "image reconstruction" instead of "image synthesis". The title can be better to emphasize on its key insight: slot-centric reconstruction. (3) the model name. GFSNet is generic, as any TTA methods that require finetuning can be named as GFSNet.

**Summary Of The Paper:**

The paper aims to tackle the out-of-domain generalization challenge for instance segmentation of 2D images and 3D point clouds. The proposed method combines the recent slot attention-based model for segmentation and the standard test-time-augmentation approach with the auxiliary reconstruction task. The proposed method significantly outperforms the prior art in the out-of-domain setting on standard benchmarks. The ablation studies show the importance of the proposed module and learning setting.

**Summary Of The Review:**

Overall, the proposed method achieves strong performance on the selected multiple datasets. However, I have concerns on (1) its novelty as it simply combines two existing ideas; (2) usability in real world as the dataset images are heavily crowded with synthetic objects, which makes it unclear if the performance gains mostly comes from the TTA framework or the segmentation backbone that is better suited for the crowded scene. The paper will be much stronger if it provides results on popular 2D image instance segmentation benchmarks.

---

> ### Author Response · Authors · 2022-11-09
> **Response to review  (part 1/2)**
>
> We are encouraged that you found our work to be an "interesting idea", have "strong performance on popular datasets" and have "detailed ablations" and "model description" . Below we address your concerns.
>
> **Q1. Lack of novelty/insight.**
>
> As Reviewer 2 points out, "this is the first work exploring the scene understanding tasks by test-time adaptation". Given that scene understanding is a central task in computer vision and given there is extensive literature on TTA for the task of image classification, we believe TTA for scene understanding is an important research direction, and our paper makes a first and a significant step in that direction.
>
> The TTA community largely focuses on different types of SSL losses, such that the gradients of the SSL loss align with the gradients from the task loss (eg. Sun et al., 2020; Bartler et al., 2022; Gandelsman et al., 2022). However, we find in our work (Table 1), specifically in the Mask2former+Recon and Mask2former+BYOL baselines, that searching only amongst different loss functions could be insufficient. It is important to look at architectural biases, specifically slot-centric biases to obtain additional improvement during TTA, especially for the task of scene understanding.
>
> Further, slot-based generative models (eg. Engelcke et al., 2019; Greff et al., 2019; Locatello et al., 2020) have not been used or developed with the forethought of test time adaptation. As a matter of fact, Engelcke et al (2020), show that TTA through reconstruction in slot-centric generative models fails, due to a reconstruction-segmentation trade-off: as the entity bottleneck gets wider, reconstruction improves but then the segmentation gets worse. In this work, we show that weak supervision can help break this trade-off and test-time adaptation using reconstruction objectives can significantly help scene decomposition.
>
> Additionally, simply mixing slot-centric architectures with TTA is not sufficient. There are additional design choices that come into play. For example, in OSRT (Sajjadi et al., 2022a) it has been found SlotMixer decoder and Broadcast decoder do equally well in unsupervised object discovery, however in our experiments (Table 5) for TTA we find the Broadcast Decoder to significantly outperform (by 11 points) the SlotMixer decoder in TTA (0.72 -> 0.83). Further finding the right set of parameters to adapt, pixel-sampling strategy, and how to supervise slot-centric models are important design choices and play an important role in this improvement (Table 5).
>
> We have provided the above-mentioned insights in Section 4.3.1 results subsection and in Introduction paragraphs 2,3 and 4.
>
> **Q2. Biased experiment setting.**
> > “the benchmark dataset (sec. 7.1-7.2) has few instances in the training set and much more instances in the test set”.
>
> Few to more instances is a very common test for strong generalization capabilities (i.e. extrapolation) [1,2,3]. Additionally, in section 7.1 we don’t just increase the number of instances but also introduce new instances not seen during training. Further in section 7.3, we show generalization to a new category Chair -> All other categories in PartNet.
>
> >“Such experiment design favors instance segmentation models for crowded scenes (i.e., slot-attention based method) over popular maskformer models.”
>
> We believe this is an incorrect/unsubstantiated claim. We are unaware of any literature that would support this claim, i.e. slot attention being particularly well suited for crowded scenes. On the contrary, MaskFormer models constitute the state of the art in supervised segmentation (which often includes crowded scenes). Could you point us to a reference that supports this claim?
>
> **References:**
>
> [1] Cai, Zhongang, et al. "Messytable: Instance association in multiple camera views." European Conference on Computer Vision. Springer, Cham, 2020.\
> [2] Chavdarova, Tatjana, et al. "Wildtrack: A multi-camera hd dataset for dense unscripted pedestrian detection." Proceedings of the IEEE Conference on Computer Vision and Pattern Recognition. 2018.\
> [3] Milan, Anton, et al. "MOT16: A benchmark for multi-object tracking." arXiv preprint arXiv:1603.00831 (2016).

---

> > ### Author Response · Authors · 2022-11-09
> > **Response to review part(2/2)**
> >
> > **Q3. Lack of results on common images (eg: COCO)**
> >
> > We have been actively working on this. This is comparatively challenging as slot-bottlenecks reduce the expressivity of the network and make it difficult to model complex datasets.
> > We think this might require us to build slot-architectures with dynamic bottlenecks such that during training there is a loose bottleneck which allows us to train supervised, while during TTA there is a tighter bottleneck such that it aligns the task of segmentation and reconstruction.  Further to go to COCO it might be important to explore different renderer architectures that can model interactions between different objects.
> >
> > To address your concern, we have run our model on a real-world salad plating dataset which we collected and will publicly release here: https://sites.google.com/view/slottta/home. As you can see, our model effectively segments the scene into objects and amodally reconstructs each one, despite heavy occlusions. For these results,  we use the single-view RGB setup presented in experimental section 4.2. We will add these results to the paper.
> >
> > **Q4. Naming issues**
> >
> > Thank you for your suggestion; we followed it. We changed "segmentation" to "instance segmentation" in most places. Additionally, we also changed our title to: "Test-time adaptation with slot-centric models" instead of "Test-time Adaptation for Segmentation via Image Synthesis". Further, we changed the model name to Slot-TTA instead of GFSNet

---

> > > ### Author Response · Authors · 2022-11-15
> > > **Results on a new distribution shift.**
> > >
> > > To satisfy your concern about biased experiment settings, we tested our model on a different distribution shift. In the test set instead of increasing the number of instances in the scene in Section 4.1, we introduced instances from new object categories. Specifically the MSN train-set consists of ShapeNet object categories (Tables, Chairs etc), whereas the new test-set consists of Google Scanned Object (GSO) [1] categories (Shoes, Stuffed toys etc). We find that our model gets a score of **0.92** before TTA and **0.95** after TTA, whereas mask2former and its TTA counterparts get a score of **0.93**, clearly demonstrating the benefit of slot-centric test-time adaptation over a state-of-the-art baseline.
> > >
> > >
> > >
> > > [1] Downs, Laura, et al. "Google Scanned Objects: A High-Quality Dataset of 3D Scanned Household Items." arXiv preprint arXiv:2204.11918 (2022).

---

> ### Author Response · Authors · 2022-11-18
> **Request for final feedback.**
>
> Dear Reviewer,
>
> We hope we have been able to address all of your concerns in the updated manuscript.
>
> As the rebuttal process ends tomorrow, please let us know if you have any further comments or queries.
>
> Thank you.

---

### Author Response · Authors · 2022-11-16
**General Response (summarizing new experiments)**

We thank all the reviewers for their feedback. We have conducted additional experiments to address reviewer concerns.  Here, we summarize the major changes below:

i) Added results for a new distribution shift experiment in Section 4.1.: instead of increasing the number of instances in the test set we sample instances from new categories, which more closely matches distribution shift scenarios encountered in practice

ii) Added results for a newly added object detection task (predicting bounding boxes) in Section 4.3.2.

iii) Added results for a real-world plating dataset (https://sites.google.com/view/slottta/home), where we show our model’s capability to do segmentation and amodal completion on highly occluded scenes.

iv) Changed paper title to “Test-time Adaptation with Slot-centric Models” and method name to “Slot-TTA” to better reflect our method.

v) Added more detailed descriptions for some terminologies.

---

### Decision · Program_Chairs · 2023-01-20

**Decision:**

Reject

**Justification For Why Not Higher Score:**

Incremental. Lacks technical novelty. Experiments are not convincing.

**Justification For Why Not Lower Score:**

N/A

**Metareview: Summary, Strengths And Weaknesses:**

in this paper, the authors proposed a test-time adaptation framework for some computer vision tasks. Though the setting studied in this paper is new, the proposed solution is incremental, and lacks technical novelty. Moreover, the experimental setting and results are not convincing.

In summary, this paper is not ready for publication in ICLR based on its current shape.